# ViDE: Tuning-Free Video Coherence via Temporal Attention Reweighting and Prompt Blending

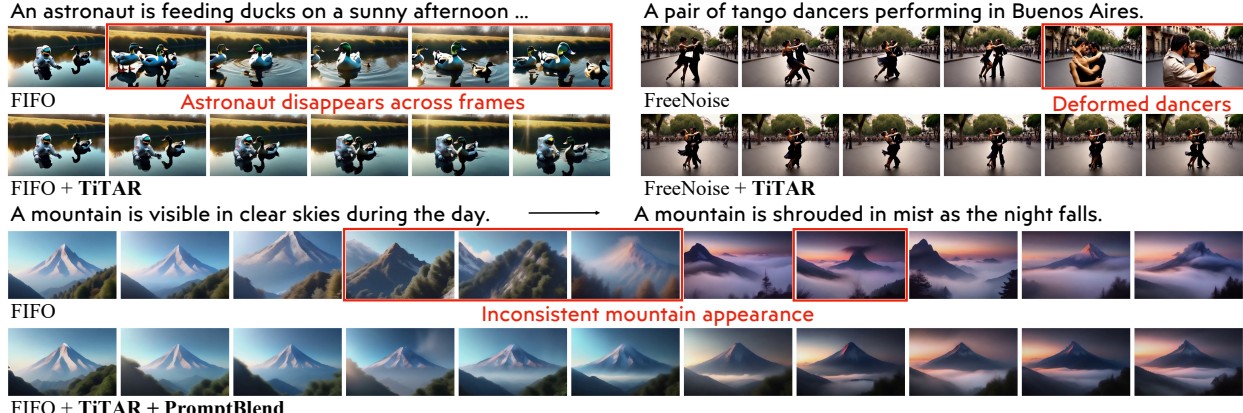

Figure 1: **Qualitative comparisons between ViDE and existing methods on long video generation**. Our ViDE framework comprises two complementary modules, TiTAR and PromptBlend, which refine video and prompt hidden states, respectively, to enhance generation quality and consistency. Applying TiTAR to single-prompt generation (top left and top right figures) markedly improves object and background consistency, while employing ViDE, *i.e.*, TiTAR and PromptBlend, for multi-prompt generation (bottom figure) further enhances overall generation quality and coherence.

## Abstract

Despite the substantial progress in long video generation, multi-prompt synthesis often encounters inconsistencies resulting from the training-inference gap caused by length extension techniques and coarse prompt interpolation. To overcome these issues, we propose the Video Diffusion with hidden states Editing (ViDE) framework, which consists of two key components. The first is the Time-frequency based Temporal Attention Reweighting (TiTAR) algorithm, which leverages the relationship between inconsistencies and diagonal elements of temporal attention. By reweighting the attention scores via the Discrete Short-Time Fourier Transform (DSTFT), TiTAR effectively reduces frame inconsistencies, a capability further corroborated by a Fourier-based analysis. The second component, PromptBlend, reduces inconsistencies in multi-prompt settings through fine-grained prompt alignment and adaptive interpolation, enabling smooth semantic transitions. Extensive experiments demonstrate the effectiveness of ViDE, demonstrating consistent and significant improvements over multiple baselines.

## 1 Introduction

The Video Diffusion Model (VDM) has achieved notable successes in a variety of tasks, including text-to-video (Chen et al., 2024; Guo et al., 2023), image-to-video (Hu, 2024), and object animation (Blattmann et al., 2023a). Among these applications, the long video generation problem has recently received increasing attention. Recent advances in VDMs have enabled the production of high-resolution videos lasting up to ten seconds (Zheng et al., 2024; Lab & etc., 2024). To further extend video duration, researchers have

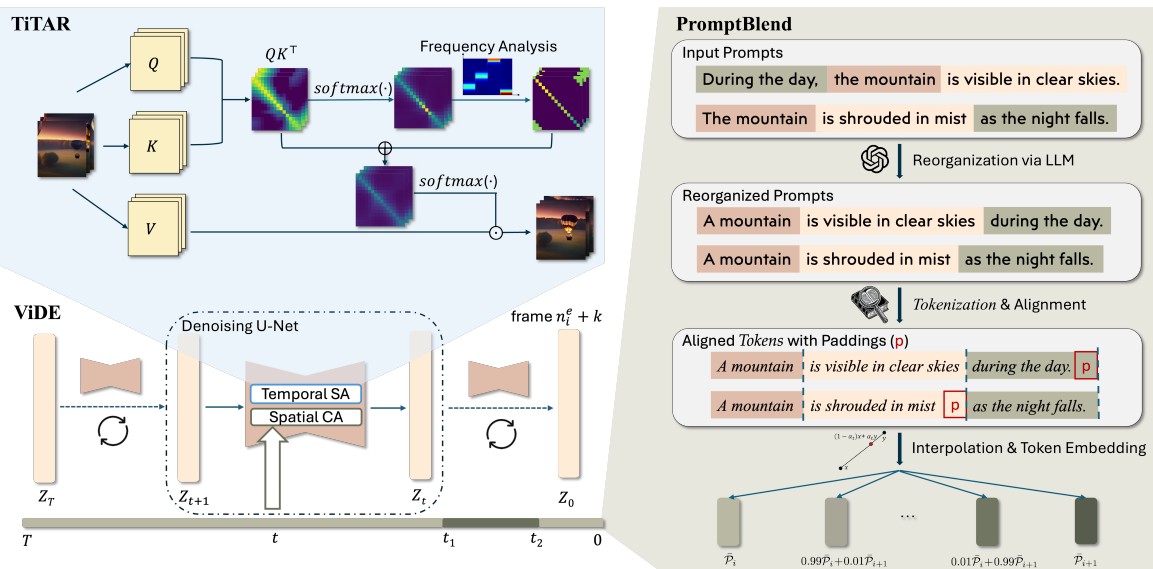

Figure 2: The Video Diffusion with Hidden States Editing (ViDE) framework comprises two complementary modules: TITAR and PROMPTBLEND. **Left:** To mitigate inconsistencies associated with excessive focus on individual frames during temporal self-attention, TITAR introduces a reweighting matrix upon the attention map at each temporal step within the denoising U-Net. The diagonal elements of this matrix are dynamically determined via frequency-based motion intensity analysis of the generated video. **Right:** For multi-prompt video generation, PROMPTBLEND improves prompt interpolation by first aligning prompts in the token space and interpolating their embeddings, then gradually blending the interpolated text conditioning into the inference process.

explored both auto-regressive pretrained models and training-free extension techniques. In parallel, prompt interpolation methods have been proposed to improve the content diversity of long videos (Kim et al., 2024a).

Despite substantial progress in long video generation, several critical challenges remain. Current length extension methods repurpose pre-trained VDMs in ways they were never originally trained for (Kim et al., 2024a; Qiu et al., 2023; Wang et al., 2023a; Lu et al., 2024). For instance, FIFO-Diffusion (Kim et al., 2024a) and FreeNoise (Qiu et al., 2023) modify the initial noise distribution to increase the number of generated frames. However, these approaches introduce a distribution gap between training and inference stages, which hampers the ability of VDMs to effectively generalize their learned priors to longer video sequences. To be specific, they often lead to *inconsistencies* in the number of subjects, background shape and color, subject appearance, and object motion patterns, as illustrated in Fig. 1 (top). Moreover, existing prompt interpolation methods typically regard the prompt tokens as a monolithic whole, ignoring their internal linguistic structure (Qiu et al., 2023). Such coarse interpolation results in unnatural and inconsistent transition frames between prompts, shown in the penultimate line of Fig. 1.

In this work, we propose Video Diffusion with Hidden States Editing (ViDE) to mitigate the abovementioned issues in long video generation. ViDE consists of two complementary modules for video- and prompt-level hidden state editing, respectively. The first module, Time-frequency based Temporal Attention Reweighting algorithm (TITAR), enhances the exploitation of prior knowledge encoded in VDMs by editing video hidden states and leveraging their intrinsic structural biases. Building on the observed correlation between inconsistencies and the diagonal values of temporal attention maps, our method attenuates these diagonal weights to improve temporal coherence. To counteract the blurriness in fast-moving regions introduced by this reweighting, we apply time-frequency analysis motivated by the Discrete Short-Time Fourier Transform (DSTFT) to estimate motion intensity across video regions and adaptively adjust diagonal attention weights, thereby mitigating blur. We formalize these intuitions with a theoretical result, representing the first performance analysis of frequency-based methods in VDMs. TITAR is a plug-and-play module that is compatible with existing length extension methods and VDM architectures.

The second module, PROMPTBLEND, is designed to align and interpolate prompt hidden states in a fine-grained manner for video consistency enhancement. PROMPTBLEND utilizes Large Language Models (LLMs) to parse prompt tokens into semantic components. It then aligns corresponding components across different prompts and performs linear interpolation on their hidden embeddings. This alignment prevents semantic confusion between unrelated components, thereby improving cross-prompt consistency. We further apply this align-then-interpolate procedure adaptively based on the denoising time index and the layer index within the network. This adaptivity exploits the coarse-to-fine nature of the diffusion process (Dieleman, 2024) to achieve more effective and precise consistency improvements.

We comprehensively evaluate ViDE on FIFO-Diffusion (Kim et al., 2024a) and FreeNoise (Qiu et al., 2023) using a diverse set of pretrained *U-Net and DiT-based* VDMs, including StreamingT2V (Henschel et al., 2024), VideoCrafter2 (Chen et al., 2024), and Open-Sora Plan (Lin et al., 2024). We further demonstrate that our method generalizes to newer architectures with *full 3D attention*, such as CogVideoX (Yang et al., 2024). Experimental results show that ViDE consistently improves the consistency of long video generation, as measured by multiple evaluation metrics from VBench and EvalCrafter in multi-prompt scenarios. In addition, we demonstrate that each component of ViDE, including TITAR, prompt alignment, and the adaptive interpolation in PROMPTBLEND, can be independently applied to enhance generation consistency. Furthermore, the robustness of the hyperparameters of ViDE is validated through extensive ablation studies.

## 2  Preliminaries

**Video Diffusion Model**   In VDMs, effectively modeling temporal correlations across frames is essential for producing coherent video sequences. Recent work (Blattmann et al., 2023b; Guo et al., 2023; Chen et al., 2024; Wang et al., 2023b; 2025) addresses this by extending pretrained text-to-image U-Net architectures with temporal attention modules, *i.e.*, 2D+1D U-Nets. The U-Net input is a latent tensor $Z \in \mathbb{R}^{C \times N \times H \times W}$, where $C$ is the channel dimension, $N$ is the number of frames, and $H$ and $W$ denote the height and width of each frame, respectively. Temporal attention is applied independently along the temporal axis for each spatial location $(h, w) \in [H] \times [W]$. For any $(h, w)$, the input to the attention module is $Q_{h,w} = [Z_{:,1,h,w}, \cdots, Z_{:,N,h,w}]^\top W_Q = Z_{:,:,h,w}^\top W_Q \in \mathbb{R}^{N \times d_k}$, $K_{h,w} = Z_{:,:,h,w}^\top W_K \in \mathbb{R}^{N \times d_k}$, $V_{h,w} = Z_{:,:,h,w}^\top W_V \in \mathbb{R}^{N \times d_v}$, and its output is:

$$\text{Att}(Q_{h,w}, K_{h,w}, V_{h,w}) = \text{sm}\left(d_k^{-1/2} Q_{h,w} K_{h,w}^\top\right) V_{h,w}, \tag{1}$$

where $\text{sm}(\cdot)$ is the row-wise softmax function, $d_k$ and $d_v$ are the embedding dimensions of keys and values.[1] Further related works are discussed in Appendix A.

Intuitively, if the attention scores are exclusively diagonal—meaning that all focus is placed on the same frame—each frame attends only to itself, leading to a temporal inconsistency issue across frames (detailed in Section 3.2.1). DiT models, such as CogVideoX (Yang et al., 2024), employ full 3D spatial-temporal attention, which aligns with our intuition and has been similarly verified.

**Fourier Analysis**   The Discrete Fourier Transform (DFT) and DSTFT are classical techniques for frequency analysis of discrete signals. They decompose a signal into a linear combination of orthogonal trigonometric basis functions. The DFT of a discrete-time signal $x : [N] \to \mathbb{R}$ of length $N$ is defined as: $\text{DFT}(x, k) = \sum_{n=0}^{N-1} x_n e^{-i \frac{2\pi k n}{N}}, \forall k \in [N]$. The DFT at frequency $k$ captures the *global* frequency content of the entire signal. However, in many scenarios, signal characteristics vary over time (Griffin & Lim, 1984; Durak & Arikan, 2003). For example, in the audio recording of a spoken sentence, each word may have distinct frequency properties. In such cases, it is more appropriate to analyze the frequency content of *local* segments of the signal. The DSTFT enables this by incorporating a *window* function $\psi : [N] \to \mathbb{R}$, yielding:

$$\text{DSTFT}(x, \psi, m, k) = \sum_{n=0}^{N-1} x_n \psi_{n-m} e^{-i \frac{2\pi k n}{N}}, \qquad \forall k, m \in [N].$$

Common choices of $\psi$ include the Hann window and the Gaussian window (Barros & Diego, 2005; Janssen,

---

[1] For simplicity, we will omit the $\sqrt{d_k}$ in (1) throughout the paper.

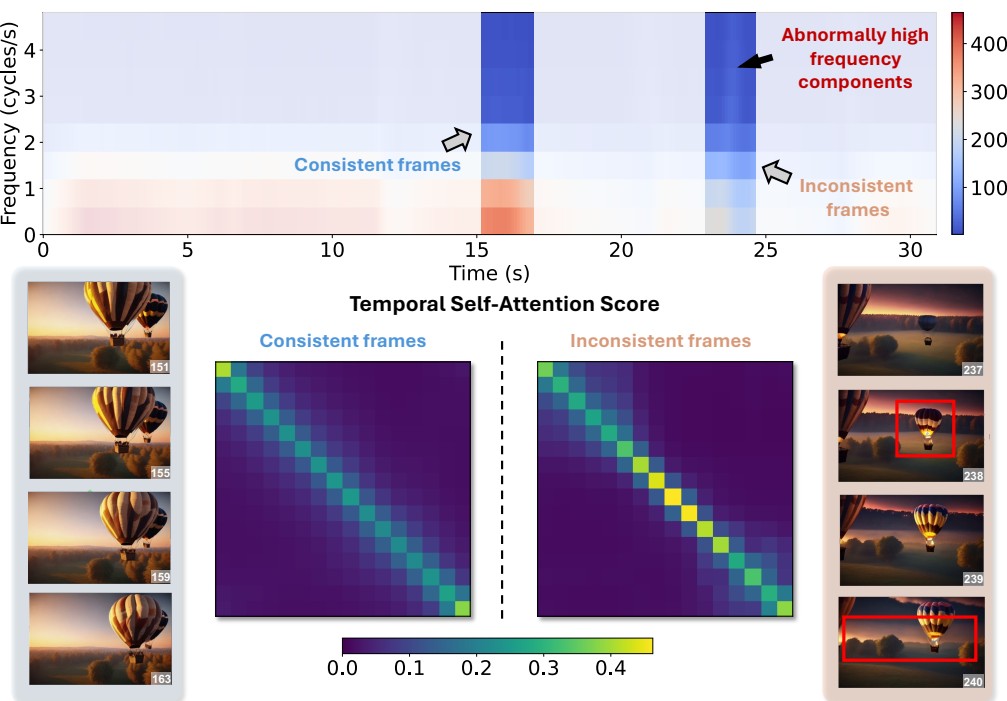

Figure 3: **Analysis of motion and temporal attention maps.** We analyze two segments within a 310-frame video as an example to illustrate the phenomenon. **Up:** The time-frequency map of the video. The left outlined segment corresponds to a well-generated sequence with consistent active motion across frames. In contrast, the right outlined segment exhibits abnormal high frequency components, capturing poorly generated content. **Middle:** The corresponding averaged temporal attention scores for these two segments. For the poorly generated frames, the attention values along the diagonal are significantly higher. **Left**: Sampled frames of video segments with normal motion. **Right**: Consecutive frames where inconsistencies occur, including the pop-up of a hot-air ballon and sudden transform of the horizon to clouds and sky. More examples can be found in Appendix J.

1991). Let $L$ denote the size of the support of $\psi$; the localized support of $\psi$ ensures that the contribution from entries outside the window centered at $m$ is suppressed, thereby enabling a localized frequency analysis. Due to the periodicity of the complex exponential $e^{-i\frac{2\pi kn}{N}}$ in $k$, lower values of $k$ (close to 0 or $N$) correspond to low frequencies, while values near $N/2$ correspond to high frequencies. This inherent periodicity also implies that the signal $x$ is treated as periodic, *i.e.*, extended such that $x_n = x_{n \bmod N}$ for all $n \in \mathbb{Z}$. In what follows, we adopt this periodic extension for all quantities related to sequence length $N$.

## 3 Methods

Section 3.1 provides an overview of our $\underline{V}$ideo $\underline{D}$iffusion with Hidden States $\underline{E}$diting (ViDE) framework, which consists of two modules: TiTAR and PromptBlend. Section 3.2 introduces TiTAR, starting from the motivating characteristics of inconsistencies, followed by the core procedure and fine-grained improvements. Section 3.3 presents the underlying intuition and implementation details of PromptBlend.

### 3.1 The ViDE Framework

As discussed in Section 1, long video generation suffers from two primary sources of inconsistency: (1) training–inference distribution gaps introduced by length extension methods, which undermines the prior learned by VDMs; and (2) coarse interpolation of multiple prompts. To address these issues, we propose ViDE (Algorithm 1), which mitigates both error sources by editing the hidden states of the denoising network.

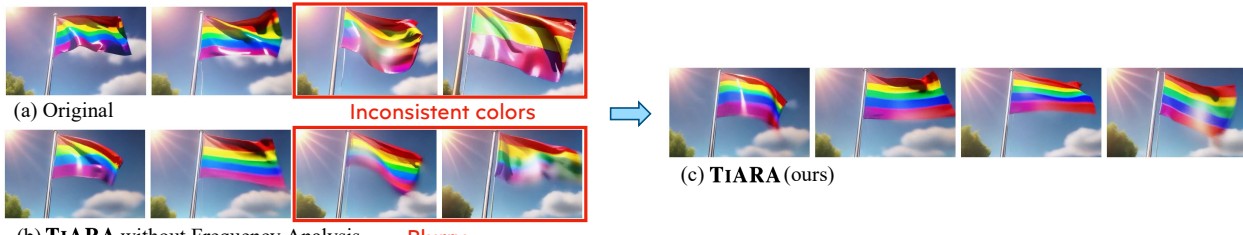

Figure 4: **Qualitative comparisons of different reweighting schemes.** (a) The original FIFO shows flag color inconsistencies, (b) The augmented versions with temporal attention reweighting only has blurred flag edges, while (c) The motion intensity adjusted reweighting—TITAR maintains clarity and consistency.

To address the first source of error, ViDE incorporates the TITAR algorithm. TITAR edits pixel-level hidden states by modifying the temporal attention layers. This operation is applied at all denoising steps (Line 8). For the second source of error, ViDE first constructs fine-grained prompt embedding interpolations via PROMPTBLEND (Line 4), and then applies them based on the video timestep, denoising step, and network layer index (Line 10). In the following sections, we will introduce TITAR and PROMPTBLEND in detail.

## 3.2 Inconsistency Mitigation via TiTAR

To mitigate inconsistencies arising from the unnatural behavior of visual hidden states, we begin by identifying their characteristics in the pixel space and the hidden state space of the network.

### 3.2.1 Hidden States and Fourier Characteristics of Inconsistent Videos

As discussed in Section 2, excessively high diagonal values in temporal attention intuitively lead to inconsistencies across video frames. We analyze a 310-frame video clip generated by FIFO-Diffusion (Kim et al., 2024a) as a representative example. Depicted in Fig. 3, the video exhibits sudden object appearances and abrupt scene transitions. Notably, the temporal attention scores in these inconsistent frames are heavily concentrated along the diagonal, in contrast to the more distributed patterns observed in consistent frames, suggesting an overemphasis on the diagonal in temporal attention may contribute to temporal incoherence.

To further investigate this correlation, we provide additional examples in Appendix J and perform a statistical analysis. With the help of visual language models (Wang et al., 2024), we select 50 consistent and 50 inconsistent video clips generated by VideoCrafter2. Fig. 5 shows the histogram of average temporal attention diagonal values. The results indicate a positive correlation between diagonal dominance and

---

**Algorithm 1** ViDE

1: **Input:** Total diffusion steps $T$, network blocks number $L$, prompts $\{P_i\}_{i=1}^m$, starting and ending points $\{(n_i^s, n_i^e)\}_{i=1}^m$, prescribed order of components $\mathcal{C} = [c_1, c_2, c_3, c_4, c_5]$, tokenizer $\mathcal{T}$, token embedder $\mathcal{E}$, frame number $n$, denoising time interval $[t_1, t_2]$, U-Net layer index $D$, frequency thresholds $\phi_1, \phi_2$, reweighting coefficient $\alpha$, base weight matrix $\Lambda \in \mathbb{R}^{N \times N}$.
2: // Prompt Embedding Preparation
3: $\mathcal{P}_C \leftarrow$ PROMPTBLEND$\big(\{P_i\}_{i=1}^m, \{(n_i^s, n_i^e)\}_{i=1}^m,$
4:     $\mathcal{C}, \mathcal{T}, \mathcal{E}, n, t_1, t_2, D\big)$
5: // Denoise Video Clips
6: **for** $n \in [n_m^e]$, $t \in [T]$, $d \in [L]$ **do**
7:     **if** Temporal attention **then**
8:         Implement TITAR$(Q, K, V, \phi_1, \phi_2, \alpha, \Lambda)$.
9:     **else if** Takes prompt embedding as inputs **then**
10:         Forward the current module with $\mathcal{P}_C(n, t, d)$.
11:     **else**
12:         Forward the current module with original inputs.
13:     **end if**
14: **end for**

---

frame-level inconsistency. This observation supports our hypothesis, namely that when temporal attention is overly concentrated along the diagonal, frames rely less on contextual information from neighboring frames, leading to increased inconsistency and reduced temporal smoothness in the generated video.

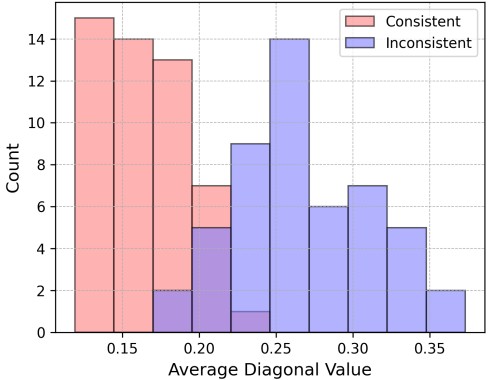 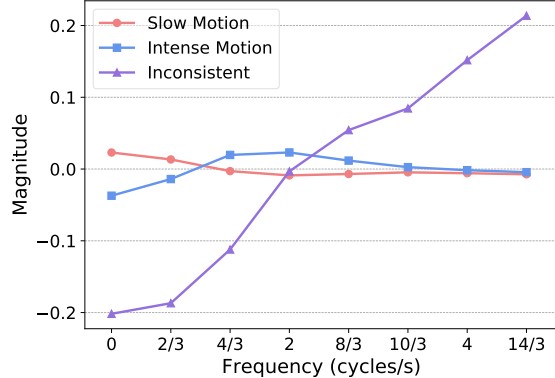

Figure 5: **Average temporal attention diagonal values.** We compare the average diagonal values of temporal attention across 50 consistent and inconsistent video clips.

Figure 6: **DSTFT of clips with different types of motions.** The y-axis shows the normalized magnitude, obtained by subtracting the mean and scaling by the maximum magnitude range, while the x-axis denotes the frequency of DSTFT.

Beyond temporal attention, we investigate additional features that may help to identify inconsistent video segments. The inconsistencies frequently manifest as abrupt changes in the subject or background, typically captured by high-frequency components of the signal. To capture this, we apply the DSTFT to video sequences to assess their frequency-domain behavior and examine the corresponding temporal attention scores during generation. As shown in Fig. 3, inconsistent segments exhibit significantly higher power in the top frequency bands of their DSTFT spectrum, while consistent segments concentrate their spectral power in lower-frequency bands. We further validate this observation through a statistical analysis presented in Fig. 6. Using a vision-language model (Wang et al., 2024), we categorize 100 video clips into three groups: *consistent clip with slow motion*, *consistent clip with intense motion*, and *inconsistent clip*. The normalized spectra, depicted in Fig. 6, reveal that inconsistent motions are typically associated with elevated high-frequency content. Motivated by this observation, we define the inconsistency power of a video segment in our theoretical analysis (Section 4) as a function of its spectral energy above a certain frequency threshold.

### 3.2.2 Temporal Attention Reweighting

Motivated by the observed correlation between temporal inconsistency and high diagonal values in the temporal attention map, we propose a temporal attention reweighting strategy to enhance video consistency. Specifically, we introduce a reweighting matrix $\Lambda = -\alpha \cdot I_{N \times N} \in \mathbb{R}^{N \times N}$, where $\alpha \geq 0$. This matrix is incorporated into the attention correlation matrix before the softmax operation in (1) to adjust the attention distribution, mitigating over-concentration on the diagonal and improving temporal consistency, as

$$\overline{\text{Att}}(Q, K, V) = \text{sm}(QK^\top + \Lambda)V = \bar{A}V. \tag{2}$$

Through subtracting positive values from the diagonal entries, we encourage each frame to draw more information from other frames and maintain stronger correlations with them, thus enhancing inter-frame consistency and smoothness. Furthermore, the weights can be optimized to balance content consistency and dynamic intensity of the video. Notably, in video generation, especially for long sequences, stronger correlations between neighboring frames are preferred over distant ones. To this end, we apply a reweighting procedure to the lower-left and upper-right corners of the temporal attention maps.

### 3.2.3 Time-frequency based Temporal Attention Reweighting

Although temporal attention reweighting effectively improves the consistency of generated videos, it is less suitable for clips with intense motion. In such cases, the weight matrix $\Lambda$ may "over-smooth" frames in the temporal axis, resulting in blurring artifacts. As illustrated in Fig. 4, FIFO-Diffusion produces inconsistent

frames with color and shape changes in the rainbow flag. Applying temporal attention reweighting restores color consistency but causes blurring along the flag's edges. This is due to the presence of high-frequency motion at the edges, which is over-smoothed by $\Lambda$ over time.

To address this issue, we introduce a time-frequency analysis approach that dynamically adjusts temporal attention reweighting. Analyzing the temporal signal in the Fourier domain enables the identification of motion intensity, which guides the assignment of values along the diagonal of the reweighting matrix $\Lambda$. Concretely, smaller (more negative) values are assigned to $\Lambda_{i,i}$ in regions with slow motion to promote information integration from neighboring frames, while values closer to zero are used in regions with intense motion, allowing frames to prioritize their own features. This adaptive strategy preserves intense motion in active areas while ensuring consistency elsewhere.

Since the temporal attention scores quantify the correlation between a frame and others, we utilize the DSTFT on each row of the normalized attention map, $A = \mathrm{sm}(Q_{h,w}K_{h,w}^{\top})$, to estimate the motion intensity. The window $\psi$ in DSTFT helps to extract local, rather than global averaged, motion intensity within the video. By applying a threshold to distinguish between high and low motion frequencies, we can obtain the high-frequency proportion of the motion signal, denoted as *motion intensity* $\rho$. Specifically, the motion intensity $\rho_i$ for $i$-th frame is defined as:

$$\rho_i := \frac{\sum_{\phi_1 \leq k < \phi_2} |\mathrm{DSTFT}(A_{i,:}, \psi, i, k)|^2}{\sum_{k < \phi_2} |\mathrm{DSTFT}(A_{i,:}, \psi, i, k)|^2}, \quad (3)$$

where $\phi_1$ is a threshold separating the intense motion from the slow motion, and $\phi_2$ is a threshold between the intense and inconsistent motion. The reweighting value $\Lambda$ should be negatively correlated with the

---

**Algorithm 2** TITAR

1: **Input:** Queries $\{Q_{h,w}\}_{h,w=1}^{H,W}$, Keys $\{K_{h,w}\}_{h,w=1}^{H,W}$,
2: Values $\{V_{h,w}\}_{h,w=1}^{H,W}$, thresholds $\phi_1, \phi_2$, reweighting
3: coefficient $\alpha$, base weight matrix $\Lambda \in \mathbb{R}^{N \times N}$.
4: **for** $h \in [H]$, $w \in [W]$ **do**
5:      $A \leftarrow \mathrm{sm}(Q_{h,w}K_{h,w}^{\top})$, $\lambda \leftarrow [\ ]$
6:      **for** $i$ in $1, ..., N$ **do**
7:          $\tilde{A}_{i,:} \leftarrow \mathrm{Pad}(A_{i,:}, \lfloor \frac{L}{2} \rfloor, \lfloor \frac{L}{2} \rfloor)$
8:          Update $\rho_i$ as in (3) with $\tilde{A}_{i,:}$
9:          $\lambda$.append($-\alpha(1 - \rho_i)$)
10:      **end for**
11:      $\mathrm{diag}(\Lambda) \leftarrow \lambda$, $Z_{h,w} \leftarrow \mathrm{sm}(Q_{h,w}K_{h,w}^{\top} + \Lambda)V_{h,w}$
12: **end for**
13: Return $\{Z_{h,w}\}_{h,w=1}^{H,W}$

---

motion intensity. In our method, we choose the relation to be linear, *i.e.*, $\Lambda_{i,i} = -\alpha(1 - \rho_i)$. The full pseudocode of our dynamic temporal attention reweighting method, TITAR, is presented in Algorithm 2. These procedures correspond to Lines 8 and 9 of Algorithm 2. We elaborate on several details of Algorithm 2. First, as noted in Section 3.2.2, the reweighting on the lower-left and the upper-right corners of the attention score is included in the base weight matrix $\Lambda$ as the input. We modify only the diagonal of $\Lambda$ according to the motion intensity. Second, for the boundary frames, we pad the attention scores in Line 7 to be consistent with the DSTFT. We adopt periodic padding in the experiments.

Although inspired by 2D+1D (spatial + temporal) U-Net-based diffusion models, TITAR is generalizable to DiT architectures that employ either 2D+1D attention (Zheng et al., 2024) and full 3D attention (Yang et al., 2024). In Section 5, we demonstrate the effectiveness of TITAR across various model variants.

### 3.3 Multi-Prompts Alignment and Interpolation

In real-world applications, generating a complete long video often requires multiple prompts to fully describe the entire event. However, inconsistencies may arise from coarse transitions between different prompts. To mitigate this issue, we propose PROMPTBLEND (Algorithm 3), an effective pipeline for multi-prompt transitions. The full algorithm is provided in Appendix B. The pipeline is shown in Fig. 2. **First**, the five components of each input prompt are organized in a prescribed order, *i.e.*, "[*The subject*] [*is doing something*] [*at some time*] [*in some place*] with [*video quality description*]". This operation can be carried out by the in-context learning of large language models (Dong et al., 2022). The prompt template is provided in Appendix C. **Then**, to align the prompts in token space, we equalize component lengths in each prompt by padding extra tokens for shorter component instances. Instead of using blank tokens for padding, we repeat tokens of the shorter instance until it shares the same length as the longest instance across prompts. We

empirically compare them in Section 5. These two procedures derive the aligned prompt embeddings $\{\bar{\mathcal{P}}_i\}_{i=1}^m$, and we then clarify how to use these embeddings as text conditioning during video generation.

Assume the total generated frames are divided into segments, marked with the starting and ending frame of prompt $i$: $\{(n_i^s, n_i^e)\}_{i=1}^m$, where indices of the transition frames are between $n_i^e$ and $n_{i+1}^s$. For the non-transition frames between $n_i^s$ and $n_i^e$, we adopt the $i$-th prompt $\mathcal{E}(\bar{\mathcal{P}}_i)$ as the text conditioning for the U-Net, where $\mathcal{E}$ represents the text encoder. For frame $n$ in the transition window $[n_i^e, n_{i+1}^s]$, we obtain the text conditioning $\mathcal{E}(\mathcal{P}_C)$ for the cross attention blocks at denoising time step $t$ and U-Net layer $d$ as

$$\mathcal{P}_C(n, t, d) = \begin{cases} (1 - a_n)\bar{\mathcal{P}}_i + a_n\bar{\mathcal{P}}_{i+1}, & t \in [t_1, t_2] \text{ or } d \geq D \\ \bar{\mathcal{P}}_i, & \text{otherwise} \end{cases},$$

and $a_n = (n - n_i^e)/(n_{i+1}^s - n_i^e) \in [0, 1]$. In practice, $[t_1, t_2]$ is set to be an interval located at the later phase of the denoising process, and $D$ is a prescribed layer index threshold, with $d \geq D$ indicating the decoder part in the U-Net. The design of this interpolation scheme is built upon two previous findings: (1) Prompt instruction at later denoising steps (*i.e.*, $t \in [t_1, t_2]$) primarily influences the generation of object shapes (Qiu et al., 2023; Balaji et al., 2022; Cao et al., 2023; Liew et al., 2022). (2) The decoder part of the denoising U-Net (*i.e.*, $d \geq D$) mainly influences the semantic details (Qiu et al., 2023) while preserving the scene layout (Cao et al., 2023). Thus, our interpolation scheme can gradually introduce elements of the next prompt to the video and is less likely to result in inconsistencies. We compare PROMPTBLEND to Motion Injection in FreeNoise (Qiu et al., 2023) in Appendix D.

## 4 Theoretical Analysis

For the theoretical analysis, we consider the case where the value $V$ of the temporal attention is the concatenation of bounded scalars, i.e., $d_v = 1$, and $|v_i| \leq B_V$ for all $i \in [N]$. The results for $d_v > 1$ can be easily derived by considering each dimension of $V$. Additional notation and definitions are provided in Appendix E. To highlight the main intuitions behind TITAR, we consider a simplification of it as follows.

$$x = \text{sm}(QK^\top)V = A^X \cdot v, \quad y = \text{sm}(QK^\top - \alpha \cdot I_{N \times N})V = A^Y \cdot v, \tag{4}$$

where $A^X$ and $A^Y$ are the attention scores of the original and improved temporal attention, respectively. The weights for all the time steps share the same value. We also simplify the weight matrix $\Lambda$ to be diagonal.

**Performance metric** Inconsistency in videos usually manifests as unnatural high-frequency changes in the frames over time, as shown in Fig. 3 and Section 3.2.1. For a pixel that varies in time, also called a *signal $x$*, we define its *inconsistency error* in a local neighborhood of time $\tau$ as

$$E(x, \tau) := \sum_{k=k_t}^{\lfloor N/2 \rfloor} |\text{DSTFT}(x, \psi, \tau, k)|, \tag{5}$$

where $|z|$ is the magnitude of the complex number $z$, and $0 < k_t \leq \lfloor N/2 \rfloor$ is the threshold of the frequency index to distinguish the inconsistent parts of the video from the consistent parts. $\phi_2$ in TITAR is an estimate of this value. For example, this can be set to $5 \cdot 2\pi/N$ in Fig. 3. The metric $E(x, \tau)$ quantifies the part of the signal that originates from rapid and unnatural motion in the video.

To facilitate our theoretical analyses, we make three assumptions. The precise statements for these assumptions are provided in Appendix F. Here, we provide abridged versions of these assumptions:

**(Existence of inconsistency)** The length-$N$ signal $x$ contains *non-negligible inconsistency error*.

**(Approximate time-homogeneity)** The attention scores $A_{i,i+k}^X$ are approximately *time-homogeneous*.

**(Separation between dynamics and inconsistency)** For frequencies contributing to the inconsistency error, the ratio of low-frequency (dynamic) power to the power of the overall signal is $\kappa < 1$.

For the third assumption, intuitively, video inconsistency often appears as temporal flickering, abrupt scene changes, or sudden changes in object appearance, which are associated with high-frequency components. As shown in Figure 6, inconsistent clips exhibit substantially greater high-frequency energy than consistent clips with either slow or intense motion. This suggests that the captured high-frequency signal reflects temporal

Table 1: **Quantitative results on single-prompt generation in FIFO-Videocrafter2 (FIFO-VC), StreamingT2V (ST2V), FreeNoise, FIFO-Open-Sora Plan (FIFO-OSP), and CogVideoX.** We only adopt length extension part of existing methods and the TiTAR component of ViDE in the single-prompt generation.

|  | SC ($\uparrow$) | BC ($\uparrow$) | TF ($\uparrow$) | MS ($\uparrow$) | CTS ($\uparrow$) | WE ($\downarrow$) | IQ ($\uparrow$) | CS ($\uparrow$) |
|---|---|---|---|---|---|---|---|---|
| FIFO-VC | 92.38 | 94.55 | 94.24 | 96.48 | 99.72 | 0.026 | 60.09 | 19.94 |
| **FIFO-VC + TiTAR** | **94.13** | **95.16** | **96.85** | **98.10** | **99.87** | **0.010** | **60.11** | **19.95** |
| ST2V | 91.08 | 95.57 | 97.33 | 98.13 | 98.67 | 0.011 | 51.49 | **18.85** |
| **ST2V + TiTAR** | **92.95** | **96.41** | **98.09** | **98.79** | **98.82** | **0.006** | **52.62** | 18.83 |
| FreeNoise | 95.48 | 97.42 | 96.50 | 97.46 | 99.87 | 0.014 | 64.64 | 20.06 |
| **FreeNoise + TiTAR** | **98.35** | **98.64** | **97.35** | **97.90** | **99.91** | **0.008** | **65.92** | **20.07** |
| FIFO-OSP | 91.06 | 95.29 | 98.10 | 98.72 | 99.96 | 0.002 | **77.91** | 18.48 |
| **FIFO-OSP + TiTAR** | **91.70** | **95.33** | **99.09** | **99.30** | **99.99** | **0.001** | 77.86 | 18.35 |
| CogVideoX | 95.51 | 95.98 | 98.36 | 98.87 | 99.93 | 0.004 | 60.89 | 18.88 |
| **CogVideoX + TiTAR** | **98.13** | **97.63** | **99.08** | **99.24** | **99.97** | **0.001** | **62.57** | **18.88** |

inconsistency rather than merely rapid motion. More validity of these three assumptions is discussed in detail in Appendix F.

**Theorem 4.1.** *Under the above assumptions, for any $\eta \in [\kappa/(1-\min_{i \in [N]} A_{i,i}^X), 1)$, there exists an $\alpha$ (see (4)) that depends on $\kappa$, $\eta$, and $A^X$, such that the inconsistency in $y = y^{(N)}$ satisfies*

$$\limsup_{N \to \infty} \frac{\mathrm{E}(y^{(N)}, \tau)}{\mathrm{E}(x^{(N)}, \tau)} \leq \eta < 1 \quad \text{for all} \quad \tau \in \mathbb{Z}.$$

The proof of Theorem 4.1 is provided in Appendix G. This result states that the inconsistency in the original signal $x$ can be reduced by the simple algorithm in (4) with a proper choice of $\alpha$. In the proof, we choose the reweighting scale as $\alpha = \log(1 + (1 - \eta)/(\eta(1 - \min_i A_{i,i}^X - \kappa)))$. A smaller value of $\kappa$ yields a larger value of $\alpha$. We deduce that it is appropriate to choose a larger $\alpha$ if $|\mathrm{DSTFT}(x, \psi, \tau, k)|$ has smaller amplitudes at higher frequencies. In Section 3.2.3, we adhere to this intuition, where we choose $\Lambda_{i,i}$ according to $|\mathrm{DSTFT}(A_{i,:}^X, \psi, \tau, k)|$. The reason to use $\mathrm{DSTFT}(A_{i,:}^X, \psi, \tau, k)$ instead of $\mathrm{DSTFT}(x, \psi, \tau, k)$ is that some videos deviate from being approximately homogeneous, and we would like to distinguish between the local frequencies at each time step.

Our proof of Theorem 4.1 exploits the interplay between transformer attention and the convolutional properties of one-dimensional signals. Specifically, we analyze in the frequency domain how masking the excessively large diagonal entries of the attention map affects the spectrum. We then relate this spectral change to the amplitude of video inconsistency defined in Eqn. (5). The theoretical findings align closely with the empirical observations reported in Section 3.2.1.

## 5 Experiments

Section 5.1 presents the experimental setup, including models, baselines, and evaluation metrics. Section 5.2 compares our approach with existing methods for single-prompt and multi-prompt generation. Finally, Section 5.3 analyzes each component of the ViDE framework and its sensitivity to hyperparameters.

### 5.1 Experimental Setting

**Implementation details.** We implement our method on five long-video generation pipelines: FIFO-Diffusion (Kim et al., 2024a) with VideoCrafter2 (Chen et al., 2024) and Open-Sora Plan (Lin et al., 2024) (denoted FIFO-VC and FIFO-OSP), FreeNoise (Qiu et al., 2023) with VideoCrafter2, StreamingT2V (Henschel et al., 2024), and CogVideoX-2B (Yang et al., 2024). Among them, CogVideoX-2B employs full 3D attention while the others use a 2D+1D structure.

Table 2: **Quantitative results on multi-prompt generation in FIFO-VC and FreeNoise.** FIFO-VC (MP) and FreeNoise (MP) refer to the multi-prompt versions from Kim et al. (2024a) and Qiu et al. (2023), respectively. Specifically, they adopt *d/swi* and *MI/interp* to transition between prompts.

| | SC (↑) | BC (↑) | TF (↑) | MS (↑) | CTS (↑) | WE (↓) | IQ (↑) | CS (↑) |
|---|---|---|---|---|---|---|---|---|
| Two-prompt Video Generation | | | | | | | | |
| FIFO-VC (MP) | 85.25 | 90.66 | 95.83 | 97.63 | 99.78 | 0.012 | **54.99** | 21.61 |
| **FIFO-VC + ViDE** | **88.87** | **92.43** | **97.68** | **98.66** | **99.89** | **0.004** | 54.79 | **22.01** |
| FreeNoise (MP) | 89.44 | 94.48 | 97.29 | 98.00 | 99.89 | 0.006 | 64.87 | **22.37** |
| **FreeNoise + ViDE** | **91.01** | **95.16** | **98.25** | **98.61** | **99.95** | **0.002** | **66.28** | 22.24 |
| Three-prompt Video Generation | | | | | | | | |
| FIFO-VC (MP) | 85.12 | 91.69 | 95.13 | 97.23 | 99.77 | 0.015 | 57.24 | 21.80 |
| **FIFO-VC + ViDE** | **88.86** | **93.53** | **97.69** | **98.47** | **99.87** | **0.005** | **59.59** | **21.98** |

For single-prompt generation, FIFO-VC, StreamingT2V and FreeNoise generate 64-frame videos, FIFO-OSP generates 317-frame videos, and CogVideoX-2B generates 48 frames. For multi-prompt generation, we generate 310 frames for two prompts and 500 frames for three prompts, with 100 transition frames between prompts. Further implementation details are provided in Appendix H.

**Prompts** We employ 34 single-prompt queries and 13 multi-prompt sets. These prompts span commonly used scenarios, varying in subject matter, background, and motion intensity. Several single-prompt queries are drawn from the list in Stream-T2V (Henschel et al., 2024), whereas all multi-prompt sets are authored by us. The complete prompt list is provided in Appendix I.

**Evaluation metrics.** We evaluate generated videos using consistency metrics from VBench (Huang et al., 2024) and EvalCrafter (Liu et al., 2024), including Subject Consistency (SC), Background Consistency (BC), Temporal Flickering (TF), Motion Smoothness (MS), Imaging Quality (IQ), Warping Error (WE), CLIP-Temp Score (CTS), and CLIP Score (CS). The definitions of these metrics are provided in Appendix H.

## 5.2 Experimental Results

Qualitative results for single-prompt and multi-prompt generation are illustrated in Fig. 1, with additional results provided in Appendix J. The videos generated using our method exhibit enhanced temporal consistency and impressive visual quality. Quantitative results are presented in Tables 1 and 2. For single prompt video generation, Table 1 indicates that TITAR significantly enhances the content consistency and temporal quality of the base models. Specifically, the improved SC, BC, and CTS imply that videos generated with TITAR enjoy greater visual and semantic consistency. Moreover, the reduced TF, WE and the increased MS underscore the effectiveness of TITAR in improving the temporal quality of generated videos. In terms of IQ, our method achieves performance comparable to the original model, indicating no degradation in the visual quality of the generated frames. The comparable CS with the original model demonstrates that our method does not compromise semantic fidelity.

For multi-prompt generation, the results in Table 2 illustrate that the complete ViDE framework, i.e., the combination of TITAR and PROMPTBLEND, achieves even greater improvements in SC and BC compared to the base model, further demonstrating its effectiveness in complex generation scenarios. We consider two baselines for prompt transition. Methods such as FIFO-Diffusion (Kim et al., 2024a) and MTVG (Oh et al., 2023) (denoted as *d/swi*) directly switch prompts during generation, which often results in noticeable scene discontinuities. Building on direct interpolation, FreeNoise (Qiu et al., 2023) further introduces a motion injection mechanism (denoted as *MI/interp*) to control transitions between prompts. However, its direct interpolation approach may inadvertently entangle distinct semantic elements of the prompts, such as blending subjects and adjectives from different sentences. In contrast, PROMPTBLEND in ViDE achieves smoother prompt interpolation through prompt alignment.

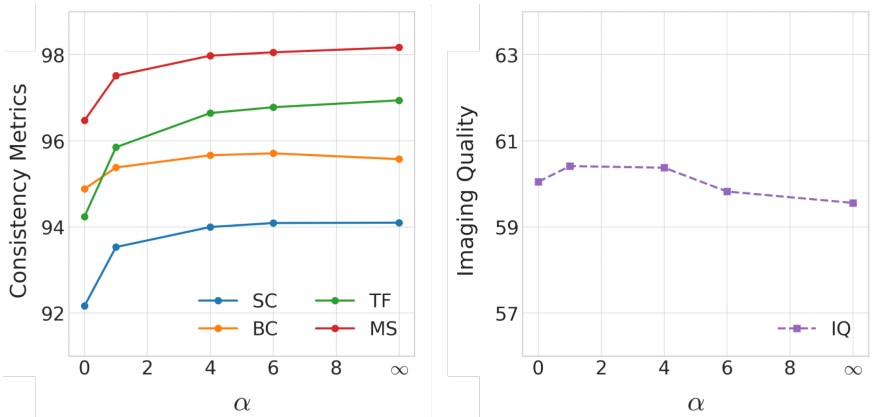

Figure 7: **Ablation study on the reweighting scale** $\alpha$. The x-axis represents the value of the reweighting scale $\alpha$ while the y-axes represent the values of consistency metrics (left) and imaging quality (right).

Table 3: **Ablation study on PromptBlend and TiTAR for multi-prompt generation with FIFO-VC.** We decompose ViDE into three components: TITAR and PROMPTBLEND, the latter consisting of prompt alignment and the adaptive application of embeddings. In the second column, we compare three options: d/swi, d/interp, and adapt/interp (Ours). Here, d/interp naively interpolates the text embeddings of the two prompts across all layers and time steps.

| Align | Interp | TITAR | SC (↑) | BC (↑) | MS (↑) |
|:---:|:---:|:---:|:---:|:---:|:---:|
| | d/swi | | 85.25 | 90.66 | 97.63 |
| ✓ | d/swi | | 85.74 | 91.17 | 97.71 |
| | d/interp | | 85.54 | 90.80 | 97.67 |
| | adapt/interp | | 86.23 | 91.00 | 97.71 |
| ✓ | d/interp | | 85.98 | 91.27 | 97.71 |
| ✓ | adapt/interp | | 86.52 | 91.36 | 97.73 |
| ✓ | adapt/interp | ✓ | **88.87** | **92.43** | **98.66** |

### 5.3 Ablation Study

In this section, we present quantitative and qualitative experiments that assess the effectiveness of each component of ViDE and its robustness to hyperparameter choices. More ablation studies can be found in Appendix J.

#### 5.3.1 Ablation of Components in ViDE

We decompose ViDE into its components for ablation: TITAR and PROMPTBLEND, with the latter further split into prompt alignment and adaptive interpolation. Results in Table 3 show consistent improvements as components are added to FIFO-VC, with the best performance achieved using both TITAR and PROMPT-BLEND. We also conduct an ablation on direct interpolation (rows 3 and 5) in Gen-L-Video (Wang et al., 2023a) (denoted as *d/interp*), where prompt embeddings are linearly interpolated and directly used as the text condition for the transition frames. This is compared against our proposed interpolation scheme in PROMPTBLEND (rows 4, 6, and 7), which incorporates temporal smoothing and prompt-aware adjustments. The results show that our interpolation strategy outperforms direct interpolation, and also works even better with prompt alignment. Furthermore, the findings underscore that prompt alignment serves as a critical component of PROMPTBLEND, consistently improving the consistency of multi-prompt video generation.

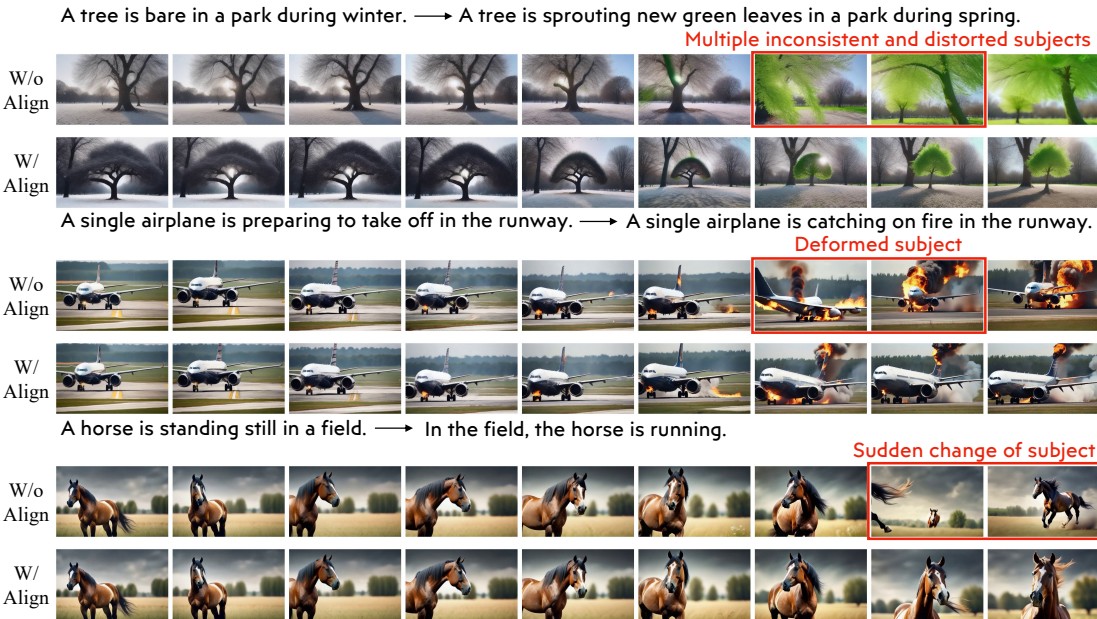

Figure 8: **Ablation study on the effect of prompt alignment on interpolation.** The experiment is conducted using FIFO and VideoCrafter2. The first row of each example is the result with interpolation but without prompt alignment; the second row is the result with both interpolation and alignment.

### 5.3.2 Ablation of reweighting scale in TiTAR

We conduct an ablation study on the reweighting scale $\alpha$ in TITAR. As shown in Fig. 7, the best performance—considering both temporal consistency and image quality—is achieved when $\alpha$ is around 4. Moreover, both the consistency metrics and the image quality remain stable across a wide range of $\alpha$ values, demonstrating the robustness of our method to this parameter.

### 5.3.3 Ablation study on the effect of prompt alignment on interpolation

We qualitatively evaluate prompt interpolation with and without our alignment technique. To isolate its effect, experiments exclude TITAR and directly use interpolated prompts for transition generation. As prompt alignment is designed to enhance the process of prompt interpolation, our analysis concentrates on the transition quality. The outcomes are presented in Fig. 8. Without prompt alignment, we observe significant degradations during transitions, which can be attributed to the lack of interpretability in the interpolated prompts. In contrast, our alignment in PROMPTBLEND produces smoother and more consistent transitions. For instance, in the first example of Fig. 8, our approach consistently preserves the tree's shape throughout all frames, whereas the baseline shows noticeable distortions during the transition from winter to spring.

## 6 Conclusion

In this work, we introduce the ViDE framework to address and enhance the consistency of long-video generation. The first component, TITAR, is inspired by empirical evidence demonstrating a correlation between video inconsistency and the diagonal elements of temporal attention matrices. By leveraging DSTFT, TITAR enables selectively and effectively mitigating inconsistencies in a nuanced manner. We further provide theoretical analysis to substantiate the efficacy of this technique. The second component, PROMPTBLEND, facilitates fine-grained prompt alignment by adaptively interpolating prompts, intuitively ensuring smooth semantic transitions throughout the video. We validate the effectiveness of the proposed ViDE framework through extensive quantitative and qualitative experiments spanning various long-video generation pipelines and prompt configurations. Comprehensive ablation studies are conducted to delineate the individual contributions of both TITAR and PROMPTBLEND.

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

# A    Related Work

## A.1    Video Diffusion Models

VDMs are an emerging class of generative models that extend the success of diffusion models from static images to video generation. The pioneering work VDMs (Ho et al., 2022) introduced temporal modeling via 3D convolutions and cross-frame attention, laying the foundation for video generation with diffusion techniques. Building on this work, a wide range of models have been proposed, including Make-A-Video (Singer et al., 2023), MagicVideo (Zhou et al., 2022), LaVie (Wang et al., 2025), and Latte (Ma et al., 2024). VideoCrafter1 (Chen et al., 2023) introduced the first open-source image-to-video (I2V) foundation model capable of generating video clips from a single image, and VideoCrafter2 (Chen et al., 2024) significantly improves generation quality with curated datasets. Open-Sora (Zheng et al., 2024) and Open-Sora-Plan (Lab & etc., 2024) are open-source efforts aimed at replicating and advancing OpenAI's Sora (OpenAI, 2024), a leading text-to-video model. More recent models, including Mochi (GenmoAI, 2024) and CogVideoX (Yang et al., 2024), tackle challenges in temporal coherence and prompt fidelity. CogVideoX leverages expert Transformers for fine-grained text control, high resolution, and strong temporal consistency, while Mochi achieves state-of-the-art performance with high-fidelity motion via its AsymmDiT architecture. Beyond diffusion, recent efforts also explore flow-based generative models for video (Reynaud et al., 2025; Jin et al., 2024).

## A.2    Long Video Generation with Diffusion Models

Despite the impressive progress of VDMs, the temporal window sizes of pretrained models are currently limited to 16–24 frames (Chen et al., 2024; Guo et al., 2023; Wang et al., 2023b). A straightforward approach to generating longer videos is to extend the temporal window, i.e., training models to denoise more frames concurrently. Open-Sora (Zheng et al., 2024) and Open-Sora-Plan (Lab & etc., 2024) are two representative efforts in this direction, capable of denoising up to 93 frames at once. However, training diffusion models with larger temporal windows demands a substantial amount of high-quality long videos, which are expensive and difficult to collect.

Another line of work explores training diffusion models with more flexible frame generation orders. In auto-regressive video diffusion models, each new chunk of frames is generated based on previously generated content. Representative examples include StreamingT2V (Henschel et al., 2024), Rolling Diffusion Model (Ruhe et al., 2024), ART-V (Weng et al., 2024), and MTVG (Oh et al., 2023). For instance, StreamingT2V introduces a conditional attention module that conditions the current chunk on selected frames from both previous and initial chunks, promoting local smoothness and global appearance consistency. A flexible history sampling strategy for training such models is proposed in Harvey et al. (2022). A complementary approach involves generating long videos by first producing a sparse set of keyframes and then interpolating between them. NUWA-XL (Yin et al., 2023), for example, is trained to predict the intermediate frame between two endpoints to generate long sequences recursively. Similarly, StoryDiffusion (Zhou et al., 2024) first generates several temporally consistent keyframes and interpolates them to produce smooth video transitions. Although these models are designed to generate longer videos, their output quality often degrades when the desired video length exceeds the training horizon, due to discrepancies between training and inference conditions.

While the aforementioned works train new diffusion models to generate long videos, a separate line of research focuses on training-free approaches. FreeNoise (Qiu et al., 2023) and FreeLong (Lu et al., 2024) extend the temporal window of a pretrained diffusion model by inputting a large number of frames simultaneously while restricting the attention window for each frame along the temporal axis. To reduce temporal inconsistency, FreeNoise also enforces that the initial noise vectors within each window are identical. Gen-L-Video (Wang et al., 2023a) generates long videos by applying multiple video diffusion models in parallel over overlapping segments; the predicted noise for each overlapping region is computed as a weighted combination of outputs from the corresponding models. FIFO-Diffusion (Kim et al., 2024a) enables long video generation in a training-free and auto-regressive manner. It organizes noisy frames into a queue and assigns higher noise magnitudes to frames based on their position. During each denoising step, the first frame in the queue is fully denoised and removed, while a new Gaussian noise frame is appended to the end. This first-in-first-out scheme allows the model to generate arbitrarily long video sequences autoregressively without retraining.

Although existing work has proposed algorithms for generating long videos, the resulting outputs often exhibit temporal inconsistencies. As illustrated in Fig. 1, baseline methods produce noticeable variations in the number, color, and shape of objects across frames. However, prior work largely are largely devoid of systematic analyses of these inconsistencies and effective strategies for mitigating them. Our study begins by empirically verifying the intuition about temporal attention discussed in Section 2.

### A.3 Frequency Analysis in Video Diffusion Models

Frequency analysis methods are widely used in the computer vision community and have also been adopted in video diffusion models to improve generation quality. For example, FreeInit (Wu et al., 2025) applies the Fourier Transform (FT) to preserve low-frequency components while progressively refining high-frequency details. FreeLong (Lu et al., 2024) also leverages FT to enhance local features while maintaining global consistency. Kim et al. (Kim et al., 2024b) demonstrate that the Wavelet Transform effectively separates information at different scales for super-resolution tasks. Additionally, frequency-domain techniques have shown promising results in generating periodic motions in videos (Li et al., 2024). However, these works typically employ frequency analysis based on heuristic or empirical intuition, without a formal theoretical foundation. In contrast, our work provides the first theoretical analysis of time-frequency methods in the context of video diffusion models, offering rigorous justification for the effectiveness.

## B PromptBlend Algorithm

---

**Algorithm 3** PROMPTBLEND

---

1: **Input:** Prompts $\{P_i\}_{i=1}^m$, starting and ending points of each prompt $\{(n_i^s, n_i^e)\}_{i=1}^m$, the prescribed order of components $\mathcal{C} = [c_1, c_2, c_3, c_4, c_5]$, tokenizer $\mathcal{T}$ and token embedder $\mathcal{E}$, frame number $n$, denoising time step interval $[t_1, t_2]$ and U-Net layer index $D$.

2:   // Prompt Embedding Alignment

3: Call LLM to organize the prompts $\{P_i\}_{i=1}^m$ into the format: $[P_{i,1}, P_{i,2}, P_{i,3}, P_{i,4}, P_{i,5}]$ according to $\mathcal{C}$.

4: **for** $k = 1, \ldots, 5$ **do**

5:     $M = \max_i \left\{ \text{length}\big(\mathcal{T}(P_{i,k})\big) \right\}_{i=1}^m$

6:     **for** $i = 1, \ldots, m$ **do**

7:       Repeat $\mathcal{T}(P_{i,k})$ until its length reaches $M$ to obtain $\overline{\mathcal{T}(P_{i,k})}$

8:     **end for**

9: **end for**

10: **for** $i = 1, \ldots, m$ **do**

11:     $\overline{\mathcal{T}(P_i)} \leftarrow \text{concat}(\{\overline{\mathcal{T}(P_{i,k})}\}_{k=1}^5)$

12:     Embed the aligned tokens $\bar{\mathcal{P}}_i \leftarrow \mathcal{E}(\overline{\mathcal{T}(P_i)})$

13: **end for**

14:   // Adaptive Application of Interpolation to Networks

15: $i \leftarrow 1$

16: **for** $n = 1, \ldots, n_m^e$ **do**

17:     **if** $n \in [n_i^e, n_{i+1}^s)$ **then**

18:       $a_n \leftarrow (n - n_i^e)/(n_{i+1}^s - n_i^e)$

19:       **if** $t \in [t_1, t_2]$ or $d \geq D$ **then**

20:         $\mathcal{P}_C(n, t, d) \leftarrow (1 - a_n)\bar{\mathcal{P}}_i + a_n \bar{\mathcal{P}}_{i+1}$

21:       **else**

22:         $\mathcal{P}_C(n, t, d) \leftarrow \mathcal{P}_i$

23:       **end if**

24:     **else if** $n = n_{i+1}^s$ **then**

25:       $\mathcal{P}_C(n, t, d) \leftarrow \mathcal{P}_{i+1}, i \leftarrow i + 1$

26:     **end if**

27: **end for**

28: **Return:** $\{\mathcal{P}_C(n, t, d)\}_{n,t,d}$

---

---

**ChatGPT Instruction**

I would like you to play the role of **the prompt organizer** that reorganize { The Prompt } into a prompt of { The Required Components }.

You **extract** all { The Required Components } from the given prompt, **reorganize** them in { The Pre-defined Order }, and **add** " $ " between different components. While doing the organization, follow the given rules:

Rule 1: If there exists **extra descriptions** about The Action, count that part into The Action, e.g. the direction of The Action, the object of The Action.

Rule 2: The output is in **the exact form** of: A/The { The Subject } $ { The Action } $ { The Place } $ { The Time } $ { Video Quality Description }.

Rule 3: If one of the components **does not exist** (e.g. The Time), return a " " for that components, e.g. A/The { The Subject } $ { The Action } $ { The Place } $ $ { Video Quality Description }.

Rule 4: Correct the **grammar** of the prompt with minimum modifications while keeping the "$"s in their positions.

The Required Components : { The Subject, The Action, The Place, The Time, Video Quality Description }.

The Prompt : { user input }

**Return only the sentence.**

---

Figure 9: **ChatGPT Instruction.**

## C  LLM Prompts for Input Prompts Organization

We use ChatGPT (i.e. GPT-4.1) to reorganize each prompt into five semantic slots: **Subject, Action, Place, Time, and Video Quality Description**. The LLM is instructed to preserve the original information, reorder it using "($)" separators, leave missing components empty, and make only minimal grammatical corrections.

In our experiments, we utilize ChatGPT for prompt organization, with exact prompt are given in Fig. 9. "The Required Components" and their sequence in the ChatGPT prompts are customizable and can be adjusted to suit the users' preferences, offering flexibility in adapting the prompts to different scenarios or tasks.

For example, "A fat rabbit wearing a purple robe walking through a fantasy landscape" is decomposed as [ A fat rabbit wearing a purple robe $ textwalking $ through a fantasy landscape $ $. ]

These five components are flexible semantic alignment slots rather than a required structure for the original prompt. Free-form prompts can be mapped to the most relevant slots, missing components remain empty, and inseparable information can stay within one component. Typical failure cases include ambiguous slot boundaries, attribute assignment, and prompts with multiple subjects or actions. Since the instruction emphasizes extraction rather than content generation, such errors usually affect slot assignment rather than the underlying semantics.

## D  Comparison to Motion Injection in FreeNoise

FreeNoise (Qiu et al., 2023) introduces Motion Injection that incorporates an interpolated prompt at later denoising time steps and U-Net decoders within the transition window. After the transition window, Motion Injection continues to apply the *first* prompt for the non-decoder components and across most time steps.

In contrast, our interpolation scheme transitions completely to the *subsequent* prompt after the transition window $[n_i^e, n_{i+1}^s]$.

Motion injection facilitates a smoother, motion-focused transition within a *single* scene, as the overall structure generation depends on the initial prompt. However, this approach is restricted to transitions between motions and has limited applicability for diverse scene changes. In PROMPTBLEND, we introduce prompt alignment to maintain component consistency during interpolation without depending on the first prompt. Consequently, PROMPTBLEND adapts seamlessly to *multiple* prompts, allowing for consistent video generation across *multiple* consecutive scenes. This is validated in our experimental results in Section 5.

## E  Additional Notations

Let $[N] := \{0, \cdots, N-1\}$. For a real number $x \in \mathbb{R}$, the largest integer that is not larger than $x$ is denoted as $\lfloor x \rfloor$. For two sequences $\{(x_i, y_i)\}_{i \in [N]}$, their discrete convolution is defined as $(x * y)_i = \sum_{j=0}^{N-1} x_j y_{i-j}$. We respectively use $I_{N \times N} \in \mathbb{R}^{N \times N}$ and $I_N \in \mathbb{R}^N$ to denote the identity matrix of size $N \times N$ and all-ones vector of size $N$. For a matrix $A \in \mathbb{R}^{m \times n}$, we use $A_{i,j}$, $A_{i,:}$, and $A_{:,j}$ to denote the $(i, j)$ component, the $i$-th row, and the $j$-th column of $A$, respectively.

## F  Assumptions

**Assumption F.1** (Existence of inconsistency). The signal $x = x^{(N)}$ of length $N$ contains non-negligible inconsistency error. More precisely, there exists a constant $C > 0$ (independent of $\tau$) such that

$$\liminf_{N \to \infty} \mathrm{E}(x^{(N)}, \tau) \geq C \text{ for all } \tau \in \mathbb{Z}.$$

This assumption states that there is a non-negligible inconsistency error in the original output of the temporal attention. Under this case, we will show that TITAR decreases $\mathrm{E}(y, \tau)$. Here we assume that such inconsistency appears in each time step $\tau \in [N]$ for the ease of analysis. Our results can be easily generalized to the situation where the inconsistency only appears in a strict subset of timesteps $N_{\text{inconsistent}} \subseteq [N] := \{0, \cdots, N-1\}$.

**Assumption F.2** (Approximate time-homogeneity). The attention scores are approximately time-homogeneous, i.e., there exists a constant $\gamma > 4$ such that

$$\left| A_{i,i+k}^X - A_{j,j+k}^X \right| = O(N^{-\gamma}) \text{ for all } i, j, k \in \mathbb{Z}.$$

This assumption states that the attention scores at different time steps are approximately homogeneous. This can be empirically observed as the similar pattern of attention scores in each row in Fig. 3. Mathematically, this assumption is satisfied when the queries and keys satisfy $q_i^\top k_j = \exp(-|i-j|)$ for all $i, j \in \mathbb{Z}$, where $|i - j|$ denotes the difference between $i$ and $j$ modulo $N$. This type of homogeneity assumption has also been adopted in existing works to analyze the properties of transformers (Dong et al., 2021).

As discussed in Section 3.2.1, large values on the diagonal of the attention score $A^X$ contribute to the inconsistency of the video. This motivates us to define the dynamic component of $A^X$ by setting the diagonal components to zeros, i.e., $A_{i,j}^{X,\text{dyn}} = A_{i,j}^X$ for $i \neq j$, and $A_{i,i}^{X,\text{dyn}} = 0$. The term "dynamic" here reflects temporal consistency and the presence of natural motion. The corresponding output is then denoted as $x^{\text{dyn}} = A^{X,\text{dyn}} \cdot v$. We term $A^{X,\text{dyn}}$ and $x^{\text{dyn}}$ as the dynamic components of $A^X$ and $x$, respectively, since $A^{X,\text{dyn}}$ and $x^{\text{dyn}}$ represent how all the other frames influence the value of the current frame. This corresponds to the *dynamics* of the video.

**Assumption F.3** (Separation between dynamic and inconsistency). For the frequencies that contribute to the inconsistency error, the low-frequency part $x^{\text{dyn}}$ has strictly less power than $x$, i.e., for all $\tau \in [N]$, and $k_t \leq k \leq \lfloor N/2 \rfloor$, we have that

$$\left| \mathrm{DSTFT}(x^{\text{dyn}}, \psi, \tau, k) \right| \leq \kappa \cdot \left| \mathrm{DSTFT}(x, \psi, \tau, k) \right| \quad \text{for} \quad \kappa < 1 - \min_{i \in [N]} A_{i,i}^X.$$

This assumption implies that the inconsistency of the dynamic part is strictly and uniformly smaller than that of the whole video. Usually, the amplitudes of DSTFT of the dynamic and smooth videos are stable, and we

can roughly regard this as a constant. Then a larger $\kappa$ implies that the DSTFT of the video $\mathrm{DSTFT}(x, \psi, \tau, k)$ has a smaller amplitude at high frequency, since $|\mathrm{DSTFT}(x, \psi, \tau, k)| = |\mathrm{DSTFT}(x^{\mathrm{dyn}}, \psi, \tau, k)|/\kappa$. Equipped with the above assumptions, we are now ready to state our main performance guarantee.

## G  Proof of Theorem 4.1

The proof of Theorem 4.1 consists of the following three steps.

- Decompose the DSTFT of signals $x$ and $y$.

- Bound each term in the decompositions.

- Conclude the proof.

**Step 1: Decompose the DSTFT of signals $x$ and $y$.**

In the whole proof, we assume that

$$A_{0,0}^X = \min_{i \in [N]} A_{i,i}^X$$

without loss of generality. Otherwise, we can replace the index 0 in the proof with $i^* = \mathrm{argmin}_{i \in [N]} A_{i,i}^X$. We would like to express the results of the attention module as follows:

$$x_i = \sum_{j=0}^{N-1} A_{i,j}^X v_j$$

$$= \sum_{j=0}^{N-1} A_{i,i+m}^X v_{i+m}$$

$$= \sum_{j=0}^{N-1} A_{0,m}^X v_{i+m} + \sum_{j=0}^{N-1} (A_{i,i+m}^X - A_{0,m}^X) v_{i+m}.$$

We define the flipped version $\bar{A}^X$ of $A^X$ as $\bar{A}_{i,j}^X = A_{i,-j}^X$. Then we have that

$$x_i = (\bar{A}_{0,:}^X * v)_i + \sum_{j=0}^{N-1} (A_{i,i+m}^X - A_{0,m}^X) v_{i+m}$$

$$= (\bar{A}_{0,:}^X * v)_i + \Delta_i^X,$$

where $*$ denotes the convolution between two signals, and $\Delta_i^X$ is the difference term. Similarly, we have that

$$y_i = (\bar{A}_{0,:}^Y * v)_i + \sum_{j=0}^{N-1} (A_{i,i+m}^Y - A_{0,m}^Y) v_{i+m}$$

$$= (\bar{A}_{0,:}^Y * v)_i + \Delta_i^Y, \tag{6}$$

$$x_i^{\mathrm{dyn}} = (\bar{A}_{0,:}^{X,\mathrm{dyn}} * v)_i + \sum_{j=0}^{N-1} (A_{i,i+m}^{X,\mathrm{dyn}} - A_{0,m}^{X,\mathrm{dyn}}) v_{i+m}$$

$$= (\bar{A}_{0,:}^{X,\mathrm{dyn}} * v)_i + \Delta_i^{X,\mathrm{dyn}},$$

where $\bar{A}^Y$ and $\bar{A}^{X,\mathrm{dyn}}$ are the flipped versions of $A^Y$ and $A^{X,\mathrm{dyn}}$, respectively. Then we would like to deduce the relationship between $A_{0,:}^X$ and $A_{0,:}^Y$. According to the definition of $A^X$ and $A^Y$, we have that

$$A_{0,i}^Y = \frac{\exp(q_0^\top k_i)}{\exp(q_0^\top k_0 - \alpha) + \sum_{j=1}^{N-1} \exp(q_0^\top k_j)}$$

$$= \frac{A_{0,i}^X}{\exp(-\alpha) \cdot A_{0,0}^X + \sum_{j=1}^{N-1} A_{0,j}^X}$$

$$= \frac{A_{0,i}^X}{1 - (1 - \exp(-\alpha)) \cdot A_{0,0}^X}$$

for $i \neq 0$, where the last equality follows from the fact that the sum over all $A_{0,i}^X$ is equal to 1. For $i = 0$, we similarly have that

$$A_{0,0}^Y = \frac{\exp(-\alpha) \cdot A_{0,0}^X}{1 - (1 - \exp(-\alpha)) \cdot A_{0,0}^X}.$$

Combining these two equations, we have that

$$A_{0,i}^Y - A_{0,i}^X = \begin{cases} \dfrac{(\exp(-\alpha) - 1)(1 - A_{0,0}^X)}{1 - (1 - \exp(-\alpha))A_{0,0}^X} \cdot A_{0,0}^X & \text{if } i = 0, \\[2ex] \dfrac{(1 - \exp(-\alpha))A_{0,0}^X}{1 - (1 - \exp(-\alpha)A_{0,i}^X)} \cdot A_{0,i}^X & \text{if } i \neq 0. \end{cases}$$

Here the equality follows from some simple algebraic manipulations. Then we decompose $A_{0,:}^Y$ as

$$\begin{aligned} A_{0,:}^Y &= \left(1 - \frac{(1 - \exp(-\alpha))(1 - A_{0,0}^X)}{1 - (1 - \exp(-\alpha))A_{0,0}^X}\right) \cdot A_{0,:}^X + \Delta^A \\ &= \frac{\exp(-\alpha)}{1 - (1 - \exp(-\alpha))A_{0,0}^X} \cdot A_{0,:}^X + \Delta^A \\ &= \iota(\alpha, A_{0,0}^X) \cdot A_{0,:}^X + \Delta^A, \end{aligned} \tag{7}$$

where $\Delta^A$ is defined as

$$\begin{aligned} \Delta^A &= \frac{1 - \exp(-\alpha)}{1 - (1 - \exp(-\alpha))A_{0,0}^X} \cdot A_{0,:}^{X,\mathrm{dyn}} \\ &= \lambda(\alpha, A_{0,0}^X) \cdot A_{0,:}^{X,\mathrm{dyn}}. \end{aligned}$$

Here $A^{X,\mathrm{dyn}}$ represents the low-frequency part of $A^X$, which is defined in Section 4. From the definition of DSTFT, we have that

$$\begin{aligned} \mathrm{DSTFT}(y, \psi, \tau, k) &= \mathrm{DFT}(y \cdot \psi_{\cdot - \tau}, k) \\ &= \big[\mathrm{DFT}(y, \cdot) * \mathrm{DFT}(\psi_{\cdot - \tau}, \cdot)\big]_k \end{aligned} \tag{8}$$

for any frequency $k \in [N]$ and shift $\tau \in [N]$, where $\psi_{\cdot - \tau}$ is the signal $\psi$ that is shifted to right by $\tau$ units. The last equality follows from the fact that DFT of the product of two signals is equal to the convolution of the DFTs of these two signals. For the first term in the right-hand side of Eqn. (8), combining Eqns. (6) and (7), we obtain

$$\begin{aligned} \mathrm{DFT}(y, k) &= \mathrm{DFT}(\bar{A}_{0,:}^Y * v, k) + \mathrm{DFT}(\Delta^Y, k) \\ &= \mathrm{DFT}(\bar{A}_{0,:}^Y, k) \cdot \mathrm{DFT}(v, k) + \mathrm{DFT}(\Delta^Y, k) \\ &= \big[\iota(\alpha, A_{0,0}^X) \cdot \mathrm{DFT}(\bar{A}_{0,:}^X, k) + \mathrm{DFT}(\bar{\Delta}^A, k)\big]\, \mathrm{DFT}(v, k) \\ &\quad + \mathrm{DFT}(\Delta^Y, k), \end{aligned} \tag{9}$$

where the second equality also follows from the fact that DFT of the convolution of two signals is equal to the multiplication of the DFTs of these two signals, and $\bar{\Delta}^A$ is the flipped version of $\Delta^A$. Combining Eqns. (8) and (9), we obtain

$$\begin{aligned} \mathrm{DSTFT}(y, \psi, \tau, k) &= \iota(\alpha, A_{0,0}^X)\big[\big(\mathrm{DFT}(\bar{A}_{0,:}^X, \cdot)\mathrm{DFT}(v, \cdot)\big) * \mathrm{DFT}(\psi_{\cdot - \tau}, \cdot)\big]_k \\ &\quad + \lambda(\alpha, A_{0,0}^X)\big[\big(\mathrm{DFT}(\bar{A}_{0,:}^{X,\mathrm{dyn}}, \cdot)\mathrm{DFT}(v, \cdot)\big) * \mathrm{DFT}(\psi_{\cdot - \tau}, \cdot)\big]_k \\ &\quad + \big[\mathrm{DFT}(\Delta^Y, \cdot) * \mathrm{DFT}(\psi_{\cdot - \tau}, \cdot)\big]_k \\ &= \iota(\alpha, A_{0,0}^X) \cdot \mathrm{DSTFT}(\bar{A}_{0,:}^X * v, \psi, \tau, k) \\ &\quad + \lambda(\alpha, A_{0,0}^X) \cdot \mathrm{DSTFT}(\bar{A}_{0,:}^{X,\mathrm{dyn}} * v, \psi, \tau, k) \end{aligned}$$

$$+ \left[ \text{DFT}(\Delta^Y, \cdot) * \text{DFT}(\psi_{\cdot - \tau}, \cdot) \right]_k, \tag{10}$$

where the first equality follows from the linearity of DFT. Similarly, we can decompose the DSTFT of $x$ and $x^D$ as

$$\text{DSTFT}(x, \psi, \tau, k) = \text{DSTFT}(\bar{A}_{0,:}^X * v, \psi, \tau, k) + \left[ \text{DFT}(\Delta^X, \cdot) * \text{DFT}(\psi_{\cdot - \tau}, \cdot) \right]_k \tag{11}$$

$$\text{DSTFT}(x^D, \psi, \tau, k) = \text{DSTFT}(\bar{A}_{0,:}^{X,\text{dyn}} * v, \psi, \tau, k) + \left[ \text{DFT}(\Delta^{X,\text{dyn}}, \cdot) * \text{DFT}(\psi_{\cdot - \tau}, \cdot) \right]_k \tag{12}$$

**Step 2: Bound each term in the decompositions.**

We would like to bound each term in the decompositions in Eqn. (10) and (11). We first bound the norms of $\text{DFT}(\Delta^X, k)$ and $\text{DFT}(\Delta^Y, k)$. For $\text{DFT}(\Delta^X, k)$, we have that

$$\left| \text{DFT}(\Delta^X, k) \right| = \left| \sum_{n=0}^{N-1} \sum_{j=0}^{N-1} (A_{n,n+m}^X - A_{0,m}^X) v_{n+m} \exp\left( -i \frac{2\pi k n}{N} \right) \right|$$

$$\leq \sum_{n=0}^{N-1} \sum_{j=0}^{N-1} |A_{n,n+m}^X - A_{0,m}^X| \cdot B_V$$

$$= O\left( \frac{B_V}{N^{\gamma-2}} \right), \tag{13}$$

where the last inequality results from Assumption F.2. Similarly, we have that

$$\left| \text{DFT}(\Delta^{X,\text{dyn}}, k) \right| = O\left( \frac{B_V}{N^{\gamma-2}} \right), \tag{14}$$

For $\text{DFT}(\Delta^Y, k)$, we first bound the difference between $A_{n,n+m}^Y$ and $A_{0,m}^Y$. From the definition of $A^Y$, for $m \neq 0$, we have that

$$A_{n,n+m}^Y - A_{0,m}^Y = \frac{A_{n,n+m}^X}{\exp(-\alpha) \cdot A_{n,n}^X + \sum_{l \neq 0} A_{n,n+l}^X}$$

$$- \frac{A_{0,m}^X}{\exp(-\alpha) \cdot A_{0,0}^X + \sum_{l \neq 0} A_{0,l}^X}$$

$$= \left( \exp(-\alpha) \cdot A_{n,n}^X + \sum_{l \neq 0} A_{n,n+l}^X \right)^{-1}$$

$$\cdot \left( \exp(-\alpha) \cdot A_{0,0}^X + \sum_{l \neq 0} A_{0,l}^X \right)^{-1} \cdot \Delta,$$

where where the second inequality follows from some simple algebraic manipulations, and the term $\Delta$ is defined as

$$\Delta = \exp(-\alpha) \cdot \left[ (A_{0,0}^X - A_{n,n}^X) A_{n,n+m}^X + (A_{n,n+m}^X - A_{0,m}^X) A_{n,n}^X \right.$$

$$\left. + \sum_{l \neq 0} A_{n,n+m}^X (A_{0,l}^X - A_{n,n+l}^X) + \sum_{l \neq 0} A_{n,n+l}^X (A_{n,n+m}^X - A_{0,m}^X) \right].$$

Using the triangle inequality, we obtain

$$|A_{n,n+m}^Y - A_{0,m}^Y| \leq O\left( \frac{\exp(2\alpha)}{N^{\gamma-1}} \right) \text{ for } m \neq 0. \tag{15}$$

For case $m = 0$, we similarly have that $|A_{n,n}^Y - A_{0,0}^Y| = O(\exp(2\alpha)N^{-\gamma+1})$. Following the same computation of Eqns. (13) and (15), we have that

$$\left| \text{DFT}(\Delta^Y, k) \right| = O\left( \frac{B_V}{N^{\gamma-3}} \right). \tag{16}$$

**Step 3: Conclude the proof.**

We conclude the proof in this final step. For the inconsistency error of the new signal $y$, we obtain

$$\sum_{k=k_{\text{thre}}}^{\lfloor N/2 \rfloor} \big| \text{DSTFT}(y, \psi, \tau, k) \big| \leq \iota(\alpha, A_{0,0}^X) \cdot \Bigg[ \sum_{k=k_{\text{thre}}}^{\lfloor N/2 \rfloor} \big| \text{DSTFT}(x, \psi, \tau, k) \big|$$

$$+ \Big| \big[ \text{DFT}(\Delta^X, \cdot) * \text{DFT}(\psi_{\cdot - \tau}, \cdot) \big]_k \Big| \Bigg]$$

$$+ \lambda(\alpha, A_{0,0}^X) \cdot \Bigg[ \sum_{k=k_{\text{thre}}}^{\lfloor N/2 \rfloor} \big| \text{DSTFT}(x^D, \psi, \tau, k) \big|$$

$$+ \Big| \big[ \text{DFT}(\Delta^{X,\text{dyn}}, \cdot) * \text{DFT}(\psi_{\cdot - \tau}, \cdot) \big]_k \Big| \Bigg]$$

$$+ \sum_{k=k_{\text{thre}}}^{\lfloor N/2 \rfloor} \Big| \big[ \text{DFT}(\Delta^Y, \cdot) * \text{DFT}(\psi_{\cdot - \tau}, \cdot) \big]_k \Big|$$

$$\leq \big( \iota(\alpha, A_{0,0}^X) + \kappa \lambda(\alpha, A_{0,0}^X) \big)$$

$$\cdot \sum_{k=k_{\text{thre}}}^{\lfloor N/2 \rfloor} \big| \text{DSTFT}(x, \psi, \tau, k) \big| + O\Big( \frac{B_V}{N^{\gamma - 4}} \Big),$$

where the first inequality follows from Eqns. (10), (11), (12) and the triangle inequality, and the second inequality follows from Assumption F.3 and Eqns. (13), (14), and (16). Since $\kappa < 1 - A_{0,0}^X$, we note that

$$\alpha := \log \frac{1 - \kappa - A_{0,0}^X \cdot \eta}{\eta(1 - A_{0,0}^X) - \kappa}$$

satisfies that $\iota(\alpha, A_{0,0}^X) + \kappa \cdot \lambda(\alpha, A_{0,0}^X) = \eta$. Since $\liminf_{N \to \infty} \text{E}(x^{(N)}, \tau) \geq C$, we conclude that the constructed $\alpha$ guarantees that

$$\sum_{k=k_{\text{thre}}}^{\lfloor N/2 \rfloor} \big| \text{DSTFT}(y, \psi, \tau, k) \big|$$

$$\leq \eta \sum_{k=k_{\text{thre}}}^{\lfloor N/2 \rfloor} \big| \text{DSTFT}(x, \psi, \tau, k) \big|$$

$$+ o\Big( \sum_{k=k_{\text{thre}}}^{\lfloor N/2 \rfloor} \big| \text{DSTFT}(x, \psi, \tau, k) \big| \Big).$$

This concludes the proof of Theorem 4.1.

# H   Additional Implementation Details

## H.1   Detailed Experimental Settings

**Pipeline configurations**   For FIFO-Diffusion, we follow the configuration of Kim et al. (2024a), where the block number is set to $n = 4$ for VideoCrafter2 and $n = 8$ for Open-Sora Plan. The DDIM sampling step is set to 64 for VideoCrafter2 and 136 for Open-Sora Plan. For FreeNoise, the window size is set to 16 with a step length of 4.

**Reweighting parameters**   The reweighting scale $\alpha$ in TiTAR is set to 5.

**DSTFT configuration** We use a Blackman window with length $L = 7$. The signal length is set to 15 when computing the DSTFT. Frequency components are categorized as follows: frequencies below 30% of the Nyquist frequency are considered low-frequency, those between 30% and 90% are mid-to-high-frequency, and those above 90% are regarded as very high-frequency components, which are associated with temporal inconsistency.

### H.2 Metrics

- Subject Consistency (SC): To assess the consistency of the subject in the video across all frames using DINO (Caron et al., 2021) feature similarity.

- Background Consistency (BC): To assess the consistency of the background across all frames using CLIP (Radford et al., 2021) feature similarity.

- Temporal Flickering (TF): To evaluate the consistency of the generated video at local and high-frequency details.

- Motion Smoothness (MS): To evaluate the quality of the movements and motions in the generated video by comparing the video interpolation result using Li et al. (2023).

- Warping Error (WE): To evaluate the temporal consistency using optical flow estimation network (Teed & Deng, 2020).

- CLIP-Temp Score (CTS): To assess the semantic consistency between each two frames by computing the cosine similarity of their CLIP (Radford et al., 2021) embeddings.

- Imaging Quality (IQ): To assess the distortion in the generated video frames, including overexposure, noise, blur, and other artifacts.

- CLIP Score (CS): To evaluate text-video consistency using CLIP model (Radford et al., 2021) as feature extractor, assess the semantic quality of the generated videos.

## I  Test Set

### I.1  Single-Prompt Test Set

The single-prompt test set is given as follow:

1. A surfer is riding a wave in a tropical beach, high quality, 4K.

2. An old man is sitting on a bench in a quiet park, high quality, 4K.

3. A scenic hot air balloon is flying in sunrise, high quality, 4K.

4. A rainbow flag is flying in the wind in a sunny day, high quality, 4K.

5. A fountain is spraying water in the center of the garden, high quality, 4K.

6. A Ferris wheel is rotating in the amusement park during twilight, high quality, 4K.

7. A book is flipping its pages on a table, high quality, 4K.

8. A colony of penguins waddling on an Antarctic ice sheet, 4K, ultra HD.

9. A pair of tango dancers performing in Buenos Aires, 4K, high resolution.

10. A red balloon is flying higher and higher in the sunlit backyard, high quality, 4K.

11. A small boat is slowly sailing across the Seine River in the sunset, high quality, 4K.

12. A lotus is floating in a tranquil pond, high quality, 4K.

13. An athlete is running on the track in the noon sunlight, high quality, 4K.

14. A fat rabbit wearing a purple robe is walking through a fantasy landscape, high quality, 4K.

15. An astronaut is feeding ducks on a sunny afternoon, reflection from the water, high quality, 4K.

16. Santa Claus is dancing.

17. People dancing in room filled with fog and colorful lights.

18. A tiger eating raw meat on the street.

19. A graceful heron stood poised near the reflecting pools of the Duomo, adding a touch of tranquility to the vibrant surroundings.

20. A woman with a camera in hand joyfully skipped along the perimeter of the Duomo, capturing the essence of the moment.

21. Beside the ancient amphitheater of Taormina, a group of friends enjoyed a leisurely picnic, taking in the breath-taking views.

22. A camel resting on the snow field.

23. A gorilla eats a banana in Central Park.

24. Time-lapse of stormclouds during thunderstorm.

25. Around the lively streets of Corso Como, a fearless urban rabbit hopped playfully, seemingly unfazed by the fashionable surroundings.

26. A musk ox grazing on beautiful wildflowers.

27. A beagle reading a paper.

28. A beagle looking in the Louvre at a painting.

29. A squirrel watches with sweet eyes into the camera.

30. Men walking in the rain.

31. A squirrel in Antarctica, on a pile of hazelnuts cinematic.

32. A young girl making selfies with her phone in a crowded street.

33. Prague, Czech Republic. Heavy rain on the street.

34. A kitten resting on a ball of wool.

### I.2 Multi-Prompt Test Set

We present the multi-prompt test set in Fig. 10.

| Multi-Prompt Test set |
|---|

Set 1: A tree is bare in a park during winter, high quality, 4K.
In the park during spring, new green leaves are sprouting on a tree, high quality, 4K.

Set 2: A forest is quiet and still in the early morning, high quality, 4K.
In the morning light, animals awaken and bring life to the forest, high quality, 4K.

Set 3: During the day, the mountain is visible under clear skies, high quality, 4K.
The mountain is shrouded in mist as the night falls, high quality, 4K.

Set 4: At sunset, a scenic hot air balloon is flying, high quality, 4K.
A single scenic hot air balloon is flying in the night, high quality, 4K.

Set 5: Under the bright sun, a mountain trail is clearly visible, high quality, 4K.
A mountain trail is shadowed and hard to see under thick clouds, high quality, 4K.

Set 6: A park is empty and still at dawn, high quality, 4K.
During the morning, the park is lively with children playing, high quality, 4K.

Set 7: A cat is sitting on a porch in the afternoon, high quality, 4K.
In the afternoon, a cat is lying down on the porch, high quality, 4K.

Set 8: Next to a tree in the park, the dog stands still, high quality, 4K.
A dog is walking away from the tree in the park, high quality, 4K.

Set 9: A horse is standing still in a field, high quality, 4K.
In the field, the horse is running, high quality, 4K.

Set 10: On the runway, a single airplane prepares for takeoff, high quality, 4K.
A single airplane is catching on fire on the runway, high quality, 4K.

Set 11: A single dark knight is riding on a black horse in the grassland, high quality, 4K.
In the grassland, a dark knight rides his black horse towards a misty forest, high quality, 4K.

Set 12: A single surfer is riding a wave on a tropical beach, high quality, 4K.
During the sunset on a tropical beach, a single surfer is riding a wave, high quality, 4K.

Set 13: In the dusk, the Seine River is gradually lighted up by streetlights, high quality, 4K.
The Seine River is occupied with boats in the dusk, high quality, 4K.

Figure 10: **Multi-prompt test set.**

Table 4: Quantitative results for single-prompt long-video generation using the VBench prompts. The baseline methods are FIFO-Videocrafter2 (FIFO-VC) and FreeNoise.

| Method | SC ↑ | BC ↑ | TF ↑ | MS ↑ | CTS ↑ | WE ↓ | IQ ↑ | CS ↑ |
|---|---|---|---|---|---|---|---|---|
| FIFO-VC | 91.17 | 93.26 | 94.60 | 96.55 | 99.77 | 0.017 | **54.61** | 20.71 |
| FIFO-VC + TɪTAR | **92.64** | **94.26** | **96.75** | **98.06** | **99.90** | **0.007** | 54.27 | **21.27** |
| FreeNoise | 96.04 | 96.71 | 96.66 | 97.41 | 99.89 | 0.012 | 59.33 | 21.08 |
| FreeNoise + TɪTAR | **98.98** | **98.59** | **97.47** | **97.88** | **99.94** | **0.006** | **61.24** | **21.11** |

Table 5: Quantitative comparison with the closely related training-free frequency-based method FreeLong on single-prompt long video generation using the VBench prompts.

| Method | SC ↑ | BC ↑ | TF ↑ | MS ↑ | CTS ↑ | WE ↓ | IQ ↑ | CS ↑ |
|---|---|---|---|---|---|---|---|---|
| FreeLong | 92.75 | 95.54 | 96.79 | 97.71 | 99.89 | 0.013 | 52.91 | 20.85 |
| TɪTAR | **98.98** | **98.59** | **97.47** | **97.88** | **99.94** | **0.006** | **61.24** | **21.11** |

## J Additional Experimental Results

**Quantitative results for single-prompt long-video generation using VBench prompts.** Table 4 reports the quantitative comparison on single-prompt long video generation using the VBench benchmark (Huang et al., 2024). We evaluate two representative long-video generation frameworks, FIFO-VideoCrafter2 (FIFO) and FreeNoise, and compare their original versions with those augmented by TɪTAR.

Across both baselines, TɪTAR consistently improves temporal consistency. On FIFO, it increases SC, BC, TF, MS, and CTS by 1.47, 1.00, 2.15, 1.51, and 0.13 points, respectively, while reducing the warp error from 0.017 to 0.007, corresponding to a 58.8% reduction. The image quality and semantic score fluctuates within a small range comparing to the baseline, indicating better temporal coherence without sacrificing text or image fidelity. Similar trends are observed on FreeNoise.

**Quantitative comparison with FreeLong.** Table 5 further compares TɪTAR with FreeLong (Lu et al., 2024), a closely related training-free method that also exploits frequency-domain information to improve long video generation. Both methods are evaluated under the same prompt set and evaluation protocol.

Compared with FreeLong, TɪTAR consistently achieves better overall performance. It improves SC, BC, TF, MS, and CTS by 6.23, 3.05, 0.68, 0.17, and 0.05 points, respectively, while reducing the warp error from 0.013 to 0.006, corresponding to a 53.8% reduction. Moreover, TɪTAR substantially improves image quality, increasing IQ from 52.91 to 61.24, and also achieves a higher cross-frame semantic score (CS) of 21.11 versus 20.85. These results highlight that merely exploiting frequency information is insufficient. By using motion-guided frequency analysis to adaptively reweight temporal attention during denoising, TɪTAR achieves substantially better temporal consistency and visual quality than the existing training-free frequency-based approach.

**Quantitative results on multi-prompt long video generation using DiTCtrl prompt set.** To further strengthen the evaluation, we additionally benchmark FIFO and FIFO+ViDE on the independently proposed DiTCtrl (Cai et al., 2025) multi-prompt prompt set under the same evaluation protocol.

Table 6 shows that ViDE consistently improves temporal consistency on this additional benchmark, and maintains a comparable imaging and semantic quality. These results demonstrate that the improvements of ViDE are not specific to our original benchmark, but generalize well to an independently constructed multi-prompt evaluation set.

**Qualitative results on single-prompt video generation** The qualitative results of single-prompt video generation based on models FIFO-VC, FIFO-OSP, FreeNoise, and StreamingT2V are displayed in

Table 6: Quantitative results on multi-prompt long video generation using DiTCtrl prompt set. We compare FIFO-VideoCrafter2 (FIFO) with and without ViDE.

| Method | SC ↑ | BC ↑ | TF ↑ | MS ↑ | CTS ↑ | WE ↓ | IQ ↑ | CS ↑ |
|---|---|---|---|---|---|---|---|---|
| FIFO-VC | 79.30 | 89.40 | 94.02 | 96.47 | 99.65 | 0.019 | **52.77** | 18.93 |
| FIFO-VC + ViDE | **82.42** | **90.32** | **96.10** | **98.10** | **99.78** | **0.007** | 52.14 | **19.03** |

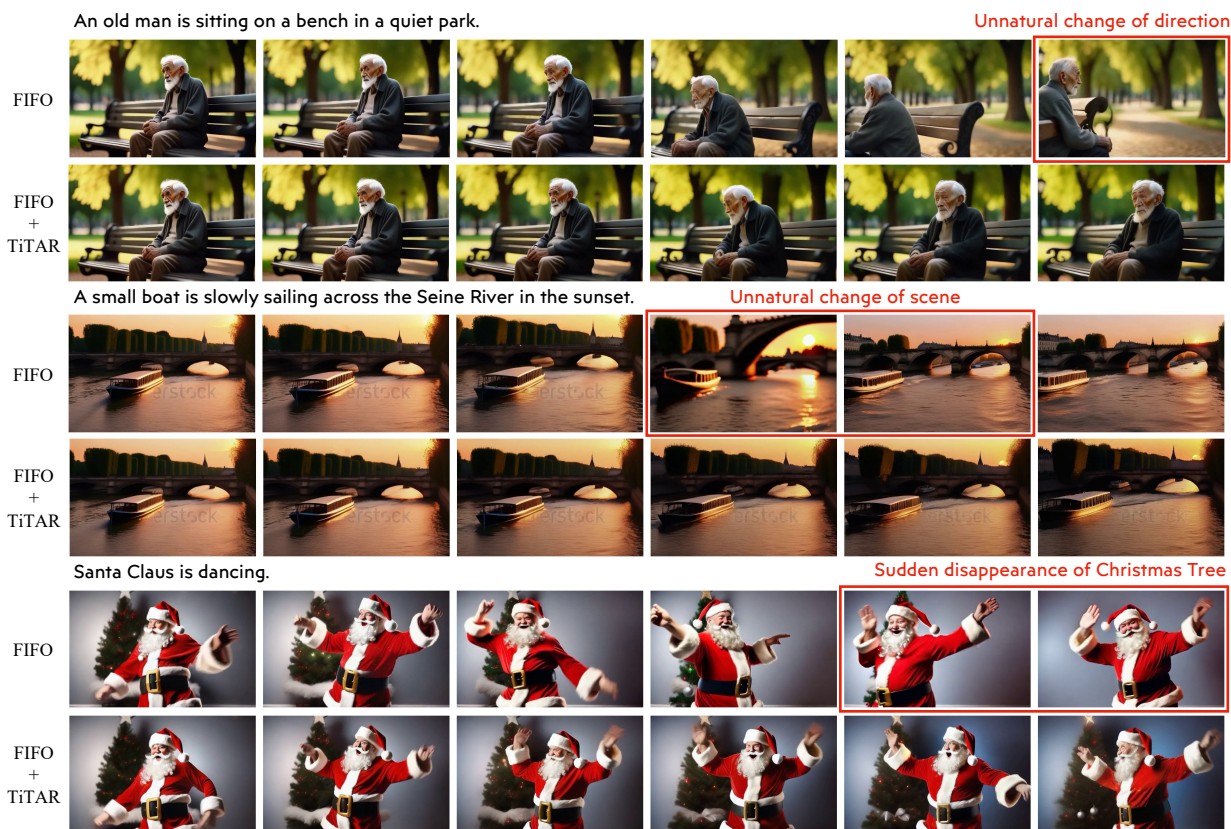

Figure 11: **Qualitative comparison on FIFO based on VideoCrafter.** The first row of each example is the result with original FIFO; the second row is the result with FIFO augmented with TITAR. The displayed frames are sampled at fixed intervals. The inconsistent parts in the videos are marked with red boxes.

Fig. 11, 16, 12, and 13, respectively. These results highlight the effectiveness of our method in addressing inconsistencies commonly observed in generated videos, such as abrupt changes in object appearance or background transitions. By applying TITAR, the generated videos exhibit enhanced temporal coherence, smoother motion, and improved overall quality, demonstrating its robustness across different models.

**Qualitative results on multi-prompt video generation** The qualitative results of multi-prompt video generation based on FIFO is presented in Fig. 17. By applying TITAR and PROMPTBLEND, inconsistencies in the generated videos are greatly reduced, achieving smoother transitions between scenes and improved overall coherence.

**More examples of time-frequency analysis** Additional time-frequency analysis is provided in Figure 18. This further verifies the conclusion in Section 3.2.1.

**Ablation study on reweighting mask shape in TiTAR** We conduct an ablation study on the mask shape of TITAR in the context of single-prompt video generation. Four cases are tested: (1) the original

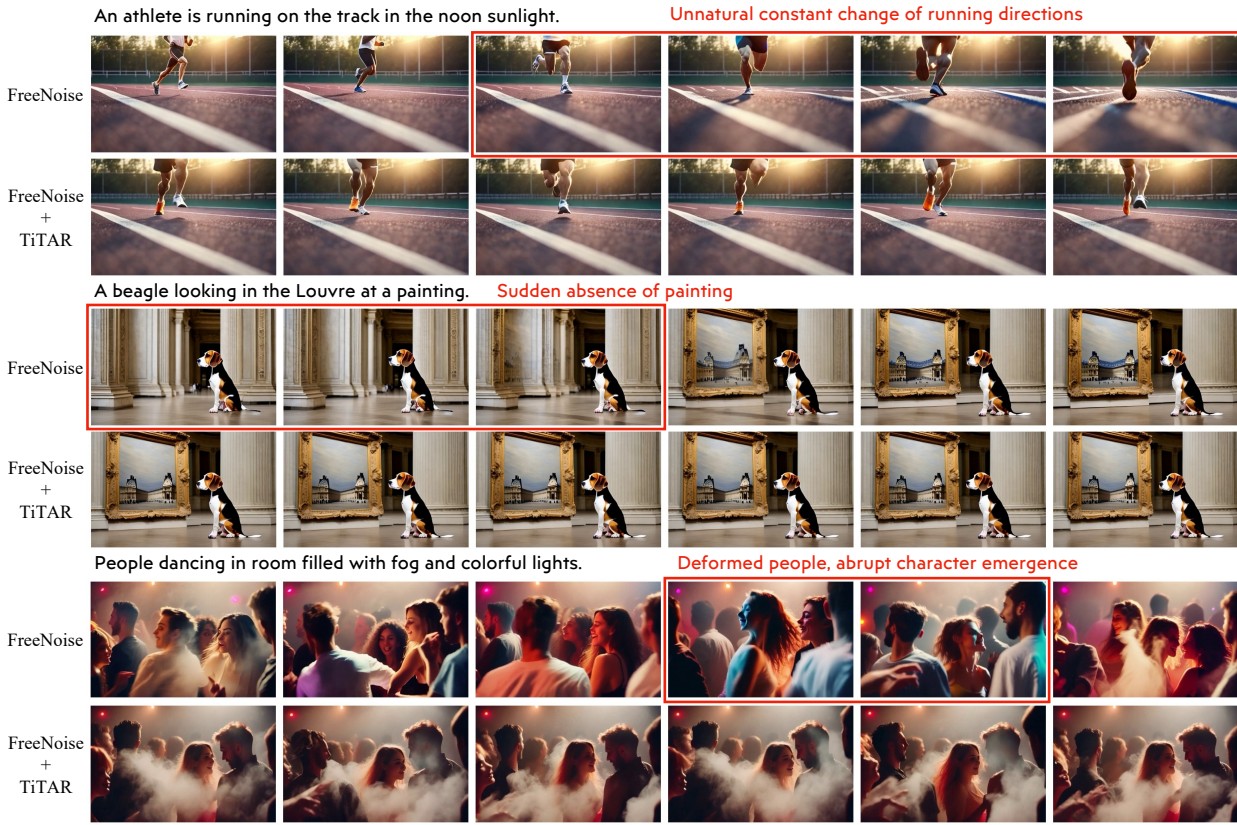

Figure 12: **Qualitative comparison on FreeNoise.** The first row of each example is the result with original FreeNoise; the second row is the result with FreeNoise augmented with TɪTAR. The displayed frames are sampled at fixed intervals. The inconsistent part in the videos are marked with red boxes.

Table 7: **Ablation study on reweighting mask shape for single-prompt generation with FIFO-VC.** "Corner" and "Diag" reweighting assign non-zero entries to Λ in (2) only at the corner and diagonal positions, respectively. In contrast, TɪTAR sets Λ with values in both regions. "No reweighting" refers to the baseline without TɪTAR.

|  | SC (↑) | BC (↑) | TF (↑) | MS (↑) | CTS (↑) | WE (↓) |
|---|---|---|---|---|---|---|
| No reweighting | 92.38 | 94.55 | 94.24 | 96.48 | 99.72 | 0.026 |
| Corner reweighting | 92.89 | 94.80 | 94.74 | 96.66 | 99.74 | 0.024 |
| Diag reweighting | 93.57 | 95.04 | 96.46 | 97.99 | 99.85 | 0.011 |
| TɪTAR | **94.13** | **95.16** | **96.85** | **98.10** | **99.87** | **0.010** |

FIFO, i.e, no reweighting; (2) reweighting applied only to the lower-left and upper-right corners of the temporal attention map; (3) reweighting applied only to the diagonal entries of the temporal attention map; and (4) TɪTAR with both reweightings. The results, shown in Table 7, indicate that applying reweighting to the corners alone provides a modest improvement, while reweighting along the diagonals yields better performance. The best results are achieved with combined reweightings. This outcome is intuitive: frames that remain consistent should assign greater weight to their immediate neighbours than to distant ones. The result underscores the contribution of each component in TɪTAR.

**Ablation study on padding scheme for PromptBlend** We conducted an ablation study comparing two strategies for prompt alignment: padding with zero tokens versus padding with repeated tokens on

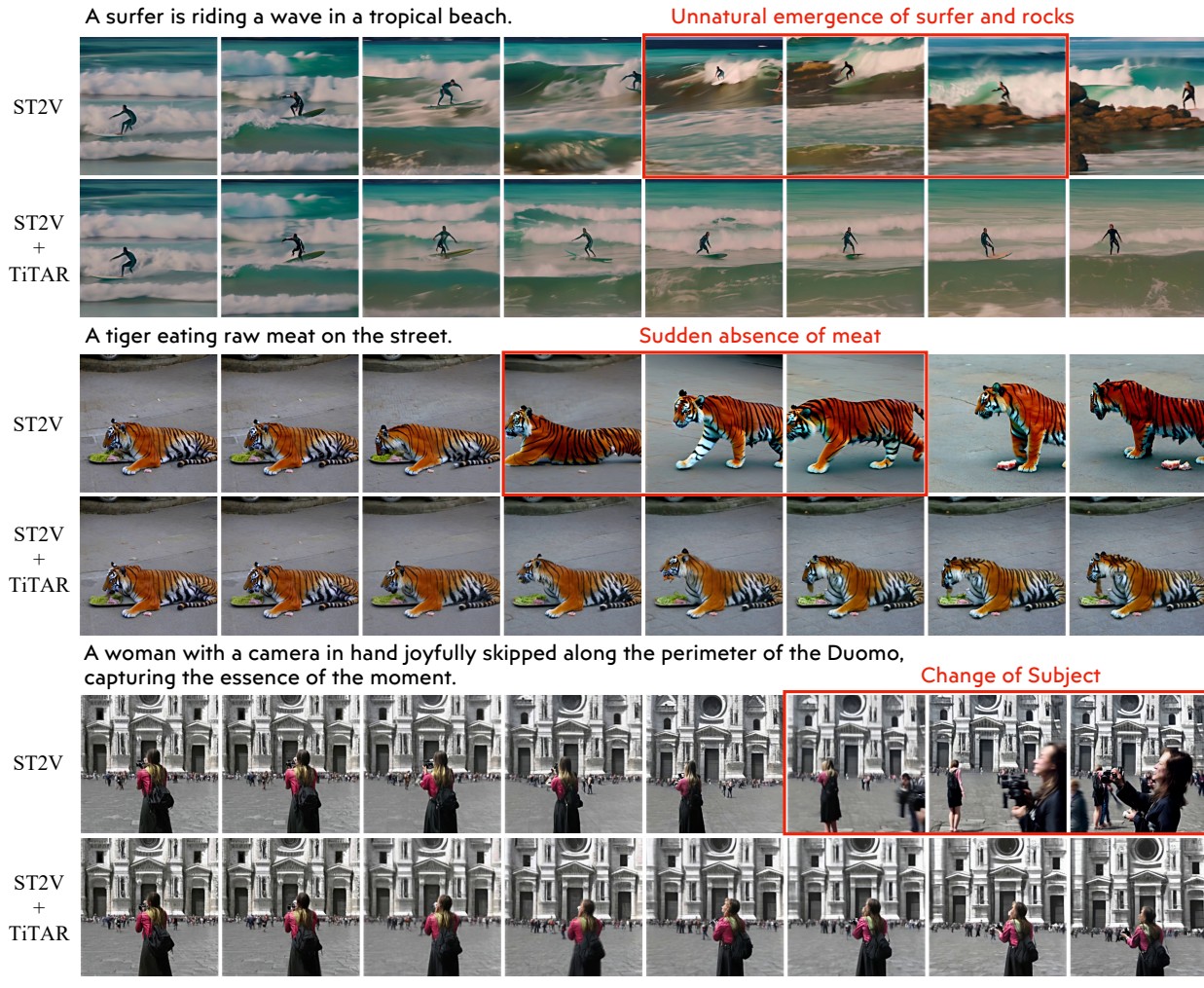

Figure 13: **Qualitative comparison on StreamingT2V (ST2V).** The first row of each example is the result with original ST2V; the second row is the result with ST2V augmented with TiTAR. The displayed frames are sampled at fixed intervals. The inconsistent part in the videos are marked with red boxes.

Table 8: **Ablation study on padding scheme for multi-prompt interpolation with FIFO-VC.** Zero padding extends the corresponding parts of the prompts to the same length using spaces, while repeat padding extends them by repeating the prompts.

|                | SC (↑) | BC (↑) | TF (↑) | MS (↑) | IQ (↑) | CS (↑) |
|----------------|--------|--------|--------|--------|--------|--------|
| Zero Padding   | 84.91  | 91.49  | 95.36  | 97.45  | 57.92  | 21.86  |
| Repeat Padding | **86.68** | **92.30** | **95.66** | **97.62** | **58.06** | **21.88** |

FIFO-VC with multi-prompt setting. The result is in Table 8. Here, zero-token padding appends a special "space" token to the end of each segment so that corresponding segments across prompts have equal length.

The results show that padding with repeated tokens consistently leads to better performance in both temporal consistency and semantic alignment. This suggests that repeating tokens helps preserve prompt-relevant information during alignment more effectively than inserting blank tokens.

Table 9: **Ablation study on frequency threshold of TiTAR with FIFO-VC.** "LF-*a*-HF-*b*" means that we set thresholds for low and high frequencies corresponding to the top *a*% and *b*% of the highest frequencies, respectively. "Original" refers to the baseline without TɪTAR.

|  | SC (↑) | BC (↑) | TF (↑) | MS (↑) | IQ (↑) |
|---|---|---|---|---|---|
| Original | 92.16 | 94.88 | 94.24 | 96.47 | 60.05 |
| TɪTAR (LF-30-HF-90) | 93.88 | 95.57 | 96.60 | 97.97 | **60.33** |
| TɪTAR (LF-50-HF-90) | 94.07 | **95.59** | 96.76 | 98.06 | 59.88 |
| TɪTAR (LF-70-HF-90) | **94.09** | 95.55 | **96.82** | **98.10** | 59.84 |

Table 10: Runtime and memory overhead of ViDE compared with the FIFO baseline. Both methods are evaluated under the same inference setting.

| Method | Runtime | Peak Host Memory |
|---|---|---|
| FIFO | 28:44.46 (1724.46 s) | 9.821 GiB |
| FIFO + TɪTAR | 29:14.66 (1754.66 s) | 9.829 GiB |
| Absolute overhead | +30.20 s | +0.008 GiB |
| Relative overhead | +1.75% | +0.08% |

**Ablation study on the frequency thresholds of TiTAR**  In Table 9, we analyze the impact of different frequency threshold settings in time-frequency analysis (TFA) in TɪTAR on video quality and consistency. The notation LF-$\kappa_1$-HF-$\kappa_2$ defines a frequency band configuration in which frequencies below $\kappa_1$ percentage are categorized as low frequency, those between $\kappa_1$ and $\kappa_2$ percentages as high frequency, and those above $\kappa_2$ percentage as abnormally high frequency. Our method assumes that regions with higher motion intensity exhibit stronger high-frequency components. Consequently, we apply less reweighting in such regions to preserve the quality of intense movements. The results show that setting $\kappa_1$ between 30 and 50 significantly improves video consistency, while maintaining or even enhancing imaging quality compared to the baseline without TɪTAR. As the high-frequency band narrows, consistency continues to improve slightly, though with a minor drop in image quality.

**Runtime and memory overhead of ViDE.**  Table 10 reports the computational overhead introduced by ViDE. Since ViDE is training-free and performs frequency analysis only along the relatively short temporal dimension, it incurs negligible additional computational cost. Compared with the FIFO baseline, ViDE increases the inference time by only 30.20 seconds (1.75%) and the peak host memory usage by merely 0.008 GiB (0.08%). These results demonstrate that the proposed temporal attention reweighting can be integrated into existing long-video generation pipelines with minimal runtime and memory overhead.

**User preference study on video consistency and overall quality**  We also conducted a user preference study to evaluate the generated single-prompt and multi-prompt videos based on human preferences. The results, shown in Fig. 14, indicate a strong preference for videos generated with our method over those generated without it, illustrating its impact on user-perceived quality. Here, our framework ViDE is applied to three baseline models for single-prompt generation and one baseline for multi-prompt generation. The details of the user study setup are in Appendix H.

**Analysis on Warping Error and Optical Flow Score**  To further evaluate the motion quality of the generated videos, we examine the relationship between two correlated factors: Warping Error (WE) and Optical Flow Score (OFS). Both metrics depend on the optical flow between consecutive frames, where WE widely used to assess temporal consistency in videos while OFS often employed to measure motion richness. In Henschel et al. (2024), a new metric called Motion Aware Warp Error (MAWE) was proposed as a ratio between WE and OFS. We take a different approach from Henschel et al. (2024) to measure the motion quality of the generated videos and real videos by analyzing the joint distribution of WE and OFS, which contains more information than MAWE. We randomly sample 170 videos each from videos generated with

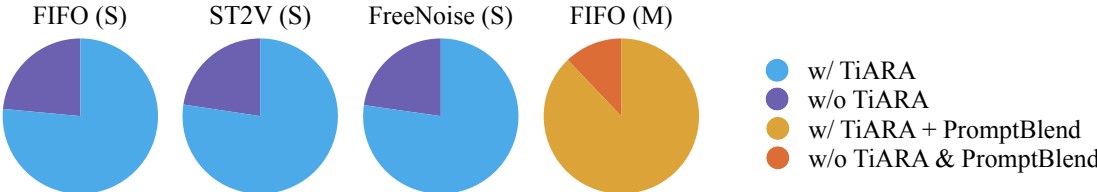

Figure 14: **User preference studies**. User preference ratio between our methods and baselines on Single- (S) and Multi- (M) prompt generation in FIFO, ST2V and FreeNoise.

FIFO, videos generated with FIFO + TɪTAR, and the real video dataset UCF-101 (Soomro, 2012). Given the differences in scale, we take the natural logarithm (ln) of both WE and OFS. To mitigate the influence of video FPS on optical flow, consecutive frames are sampled at approximately 0.1 seconds for all videos, equivalent to an effective FPS of 10. The result is plotted in Fig. 15. We state our conclusions from two perspectives.

**Firstly,** as depicted in the graph, the relationship between $\ln(\text{WE})$ and $\ln(\text{OFS})$ can be approximated by linear regression. Notably, the regression slope for our method is lower than that of the base model FIFO and closer to the slope observed in real videos. This suggests that as the motion intensity increases in the generated videos, our method better controls WE, resulting in smoother motion and better alignment with the real videos. **In addition**, we examine the similarity of the joint distribution of $\ln(\text{WE})$ and $\ln(\text{OFS})$. To do this, we first estimate the joint distributions using Kernel Density Estimation with a Gaussian kernel applied to the 170 data points for each group. Specifically, the distributions are modeled using the uniformly-weighted Gaussian Mixture Models with means determined by the samples and the bandwidth obtained from the Scott's Rule. This can be done via the `gaussian_kde` package in Python. Then, the Kullback–Leibler (KL)-divergence is computed, which measures the similarity between the distributions. The KL-divergence between the $\ln(\text{WE})$ and $\ln(\text{OFS})$ distributions of our videos and real videos is 0.82, whereas the KL-divergence for FIFO-generated videos and real videos is 1.13. This reduction in divergence indicates that videos generated with TɪTAR exhibit motion patterns that are more similar to those in real videos, further validating the effectiveness of our approach in achieving realistic motion quality.

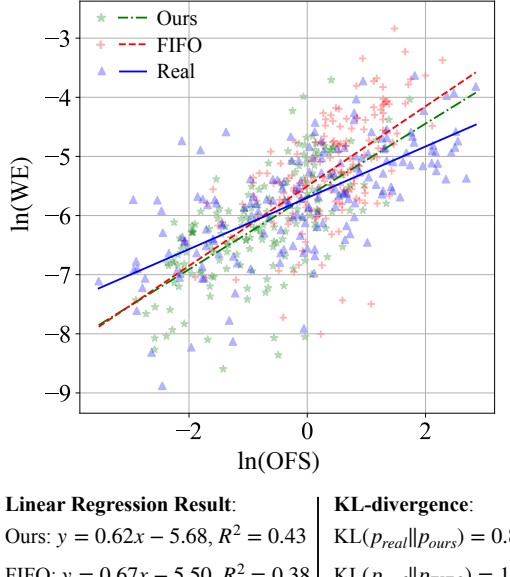

**Linear Regression Result**:

Ours: $y = 0.62x - 5.68$, $R^2 = 0.43$

FIFO: $y = 0.67x - 5.50$, $R^2 = 0.38$

Real: $y = 0.43x - 5.70$, $R^2 = 0.46$

**KL-divergence**:

$\text{KL}(p_{real}\|p_{ours}) = 0.82$

$\text{KL}(p_{real}\|p_{FIFO}) = 1.13$

Figure 15: **Linear regression of WE and OFS.**

We conclude that methods equipped with TɪTAR can generate motions that are more aligned with the real-world videos than the baselines.

## K    Limitation

This work focuses on mitigating the inconsistency issue present in current long-video generation models in a training-free manner. While our method demonstrates improved video quality, its ability to model real-world physical laws remains largely limited by the underlying base model. To further enhance this capability, we believe that incorporating training is necessary.

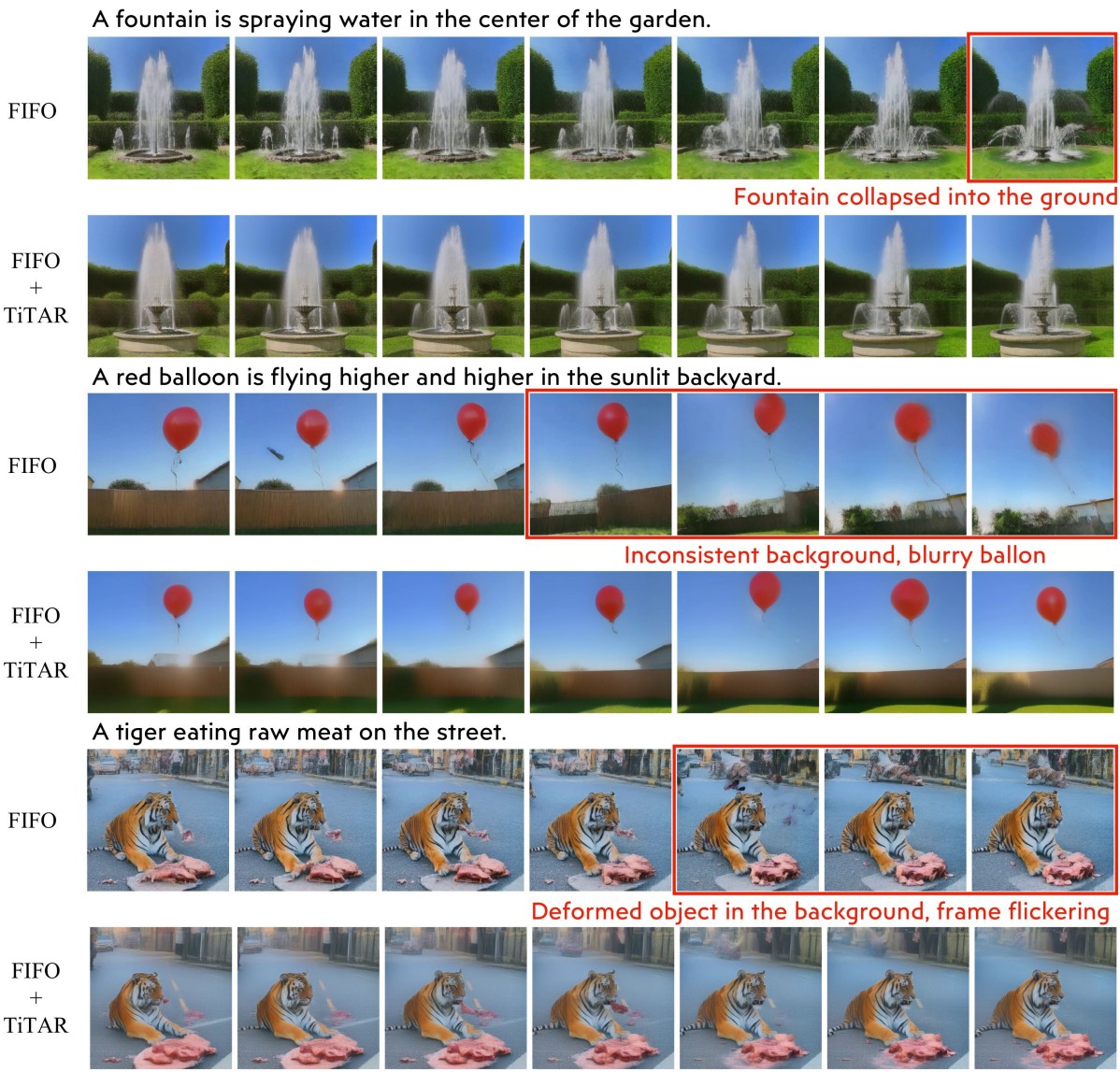

Figure 16: **Qualitative comparison on FIFO based on Open-Sora Plan.** The first row of each example is the result with original FIFO; the second row is the result with FIFO augmented with TɪTAR. The displayed frames are sampled at fixed intervals. The inconsistent part in the videos are marked with red boxes.

### K.1    Possible Negative Societal Impacts

While our proposed methods TɪTAR and PʀOMPTBLEND improve the consistency and quality of video generation using diffusion models, they also present potential risks of misuse. Enhanced video generation techniques could be exploited to create realistic but deceptive content, such as deepfakes or misleading media, with harmful societal consequences, including the spread of misinformation or manipulation of public opinion. Additionally, the ability to generate high-quality, seamless videos may pose privacy risks if applied to sensitive or unauthorized data. These concerns highlight the importance of ethical considerations and responsible usage of video generation technologies.

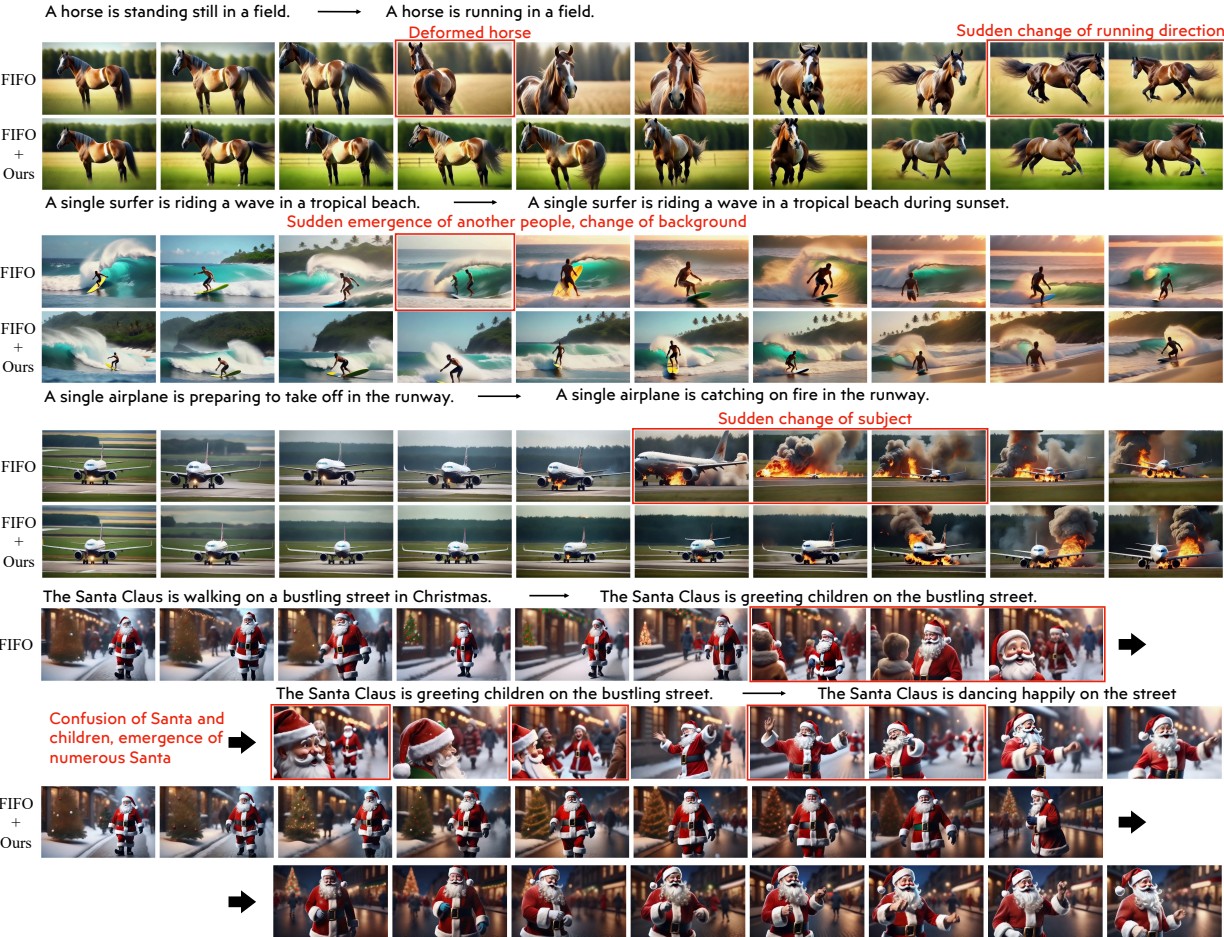

Figure 17: **Qualitative comparison on multi-prompt video generation.** The first two examples are generated with two prompts, and the third example is generated with three prompts. The first row of each example is the result with original multi-prompt FIFO; the second row is the result with FIFO + TITAR + PROMPTBLEND. The displayed frames are sampled at fixed intervals. The inconsistent part in the videos are marked with red boxes.

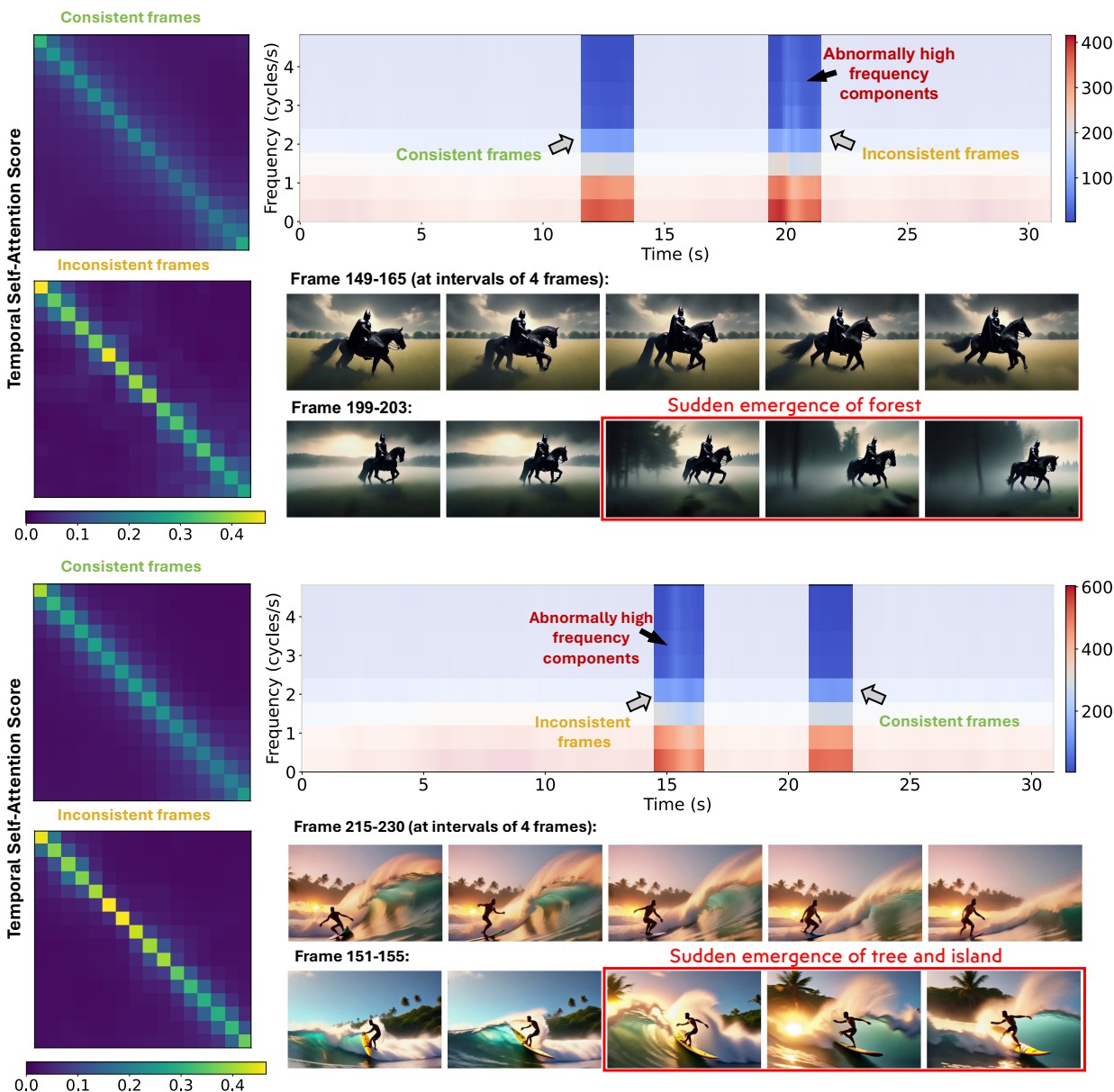

Figure 18: **More examples on the frequency analysis and attention map.**

