# OpenReview forum: "ViDE: Tuning-Free Video Coherence via Temporal Attention Reweighting and Prompt Blending"
_TMLR — Under review for TMLR_

### Review · Reviewer_zTcy · 2026-05-18

**Summary Of Contributions:**

The paper proposes ViDE, a tuning-free framework for improving temporal coherence in long video generation. It focuses on two practical issues in current video generation pipelines: frame-level inconsistencies caused by length-extension methods, and abrupt transitions in multi-prompt generation.

ViDE has two components. TiTAR modifies temporal attention by down-weighting diagonal entries, encouraging frames to attend more to neighboring frames rather than mostly to themselves. It also uses a DSTFT-based motion-intensity estimate to reduce the risk of over-smoothing highly dynamic regions. PromptBlend targets multi-prompt transitions by decomposing prompts into semantic components with an LLM, aligning them in token space, and adaptively interpolating them across denoising steps and U-Net layers.

The paper's main appeal is that it proposes a simple inference-time intervention for a real problem in long video generation. The method is easy to insert into existing pipelines, and the experiments show improvements on several automatic temporal-consistency metrics. The ablations are also useful for understanding the roles of TiTAR and PromptBlend. I found the diagonal-attention observation particularly interesting, since it gives a concrete and intuitive motivation for the proposed reweighting.

My main concerns are about the completeness of the evidence rather than the relevance of the problem. The paper does not compare against FreeLong or another very close training-free, frequency-based baseline, which makes the empirical positioning incomplete. The evaluation is also relatively small for the multi-prompt setting, relying on only 13 author-written examples. Runtime and memory overhead are not reported, even though TiTAR seems likely to add non-trivial computation. Finally, PromptBlend's reliance on LLM-based parsing is not analyzed in enough detail, and the paper sometimes describes the role of diagonal attention in a more causal way than the current correlation-based evidence can support.

**Audience:**

Yes

**Audience Explanation:**

Long video generation and temporal coherence are active topics in generative modeling, and the paper studies a practical inference-time approach that can be applied to existing video diffusion pipelines. Even though I have concerns about the strength of some claims and the completeness of the evaluation, the main observation about temporal attention and the proposed tuning-free intervention would likely be of interest to researchers working on video generation, diffusion models, and efficient adaptation methods.

**Broader Impact Concerns:**

I do not see immediate ethical concerns beyond those generally associated with video generation models. However, since the proposed method improves temporal coherence and can make generated videos appear more realistic, it may increase risks related to deceptive synthetic media, deepfakes, and disinformation. I recommend that the authors add a brief broader impact discussion covering these risks and possible mitigation strategies, such as watermarking, provenance tracking, and responsible deployment safeguards.

**Claims And Evidence:**

Yes

**Claims Explanation:**

Yes, for the most part. The paper supports its main empirical claims with improvements on automatic temporal-consistency metrics across multiple video generation pipelines, and the ablation studies provide useful evidence for the contribution of TiTAR and PromptBlend. That said, I do not think the evidence is equally strong for all aspects of the paper. The empirical comparison is incomplete without FreeLong or a similarly close training-free, frequency-based baseline, the multi-prompt evaluation is quite small, and runtime/memory overhead is not reported even though TiTAR introduces additional DSTFT operations. The paper would also benefit from more analysis of PromptBlend's LLM-based decomposition and a more cautious interpretation of the diagonal-attention correlation. Overall, the core empirical findings are supported, but the evidence would be substantially more convincing with a closer baseline comparison, larger standardized prompt evaluation, runtime/memory reporting, and possibly a small human evaluation.

**Requested Changes:**

I would consider the following changes critical to securing my recommendation for acceptance:

1. Add a direct comparison with FreeLong or another closely related training-free, frequency-based long-video generation method. FreeLong is cited in the related work and appears to be one of the closest prior methods in terms of problem setting and methodology. Without this comparison, it is difficult to assess whether TiTAR provides a meaningful advantage over existing frequency-based approaches.

2. Report runtime and memory overhead. TiTAR applies DSTFT operations across attention rows, spatial locations, temporal attention layers, and denoising steps, so the computational cost may be non-trivial. Since the method is presented as plug-and-play, the paper should report wall-clock time and peak memory overhead, ideally for both short- and long-video settings.

3. Strengthen the evaluation protocol with a larger and more standardized prompt set, especially for multi-prompt generation. The current 13 author-written multi-prompt examples are too limited to fully support claims about transition quality. Using prompts from an existing benchmark or an independently constructed prompt set would reduce concerns about selection bias.

The following changes would also strengthen the paper, although I would not view each of them as individually critical:

1. Add a small human evaluation focused on temporal coherence, motion naturalness, and transition quality. This would complement the automatic VBench and EvalCrafter metrics, which may not fully capture human perception of video quality.

2. Provide more details and analysis for PromptBlend's LLM-based decomposition. This should include which LLM is used, the cost per prompt, examples of failure cases, and how the method behaves when prompts do not fit the five-component template.

3. Soften the causal language around diagonal attention. The current evidence shows a correlation between diagonal dominance and inconsistency, but does not by itself establish that diagonal attention is the cause of inconsistency.

---

> ### Author Response · Authors · 2026-06-28
>
> We thank the reviewer for highlighting the practical relevance of the problem, the usefulness of the diagonal-attention observation, and the value of a tuning-free intervention. We address each concern below.
>
> ### 1. A direct comparison with FreeLong is missing
>
> Thanks for the suggestions. We add a comparison with Freelong under the same evaluation settings as follows, where we measure the **consistency** (Subject Consistency (SC), Background Consistency (BC), Temporal Flickering (TF), Motion Smoothness (MS), CLIP-Temp Score (CTS), Warping Error (WE)) ,and **imaging and semantic quality** (Imaging Quality (IQ), and CLIP Score (CS)):
>
> | Method   |      SC ↑ |      BC ↑ |      TF ↑ |      MS ↑ |     CTS ↑ |      WE ↓ |      IQ ↑ |      CS ↑ |
> | -------- | --------: | --------: | --------: | --------: | --------: | --------: | --------: | --------: |
> | FreeLong |     92.75 |     95.54 |     96.79 |     97.71 |     99.89 |     1.276 |     52.91 |     20.85 |
> | Ours     | **98.98** | **98.59** | **97.47** | **97.88** | **99.94** | **0.589** | **61.24** | **21.11** |
>
>
> These additional experiments **further validate the effectiveness** of our method compared with existing methods. They provide a more standardized evaluation, and better position TiTAR relative to existing training-free consistency methods.
>
> ### 2. Runtime and memory overhead are not reported
>
> We agree that computational overhead is an important consideration for a plug-and-play inference-time method. We therefore measure it by reporting wall-clock runtime and peak GPU memory under identical hardware and generation settings.
>
> | Method | Runtime | Peak host memory |
> |---|---:|---:|
> | FIFO | 28:44.46 (1724.46 s) | 9.821 GiB |
> | FIFO + ViDE | 29:14.66 (1754.66 s) | 9.829 GiB |
> | Absolute overhead | +30.20 s | +0.008 GiB |
> | Relative overhead | +1.75% | +0.08% |
>
> The results suggest ViDE requires **negligible** additional runtime and memory cost. This is because TiTAR does not introduce trainable parameters, and its DSTFT operations are performed primarily along the relatively short temporal dimension. Nevertheless, we agree that measured costs are more informative than a complexity argument alone and will include them in the revision.
>
> ### 3. The multi-prompt evaluation is too small and may contain selection bias
>
> The original 13 prompt groups were intended as **controlled diagnostic cases** rather than a comprehensive benchmark. To provide a larger and independently constructed evaluation, we **additionally test on the DiTCtrl[1] multi-prompt set**:
>
> | Method      |  SC ↑ |  BC ↑ |  TF ↑ |  MS ↑ | CTS ↑ |  WE ↓ |  IQ ↑ |  CS ↑ |
> | ----------- | ----: | ----: | ----: | ----: | ----: | ----: | ----: | ----: |
> | FIFO        | 79.30 | 89.40 | 94.02 | 96.47 | 99.65 | 0.019 | **52.77** | 18.93 |
> | FIFO + Ours | **82.42** | **90.32** | **96.10** | **98.10** | **99.78** | **0.007** | 52.14 | **19.03** |
>
> We also report results on the official VBench prompts for the single-prompt setting as an additional standardized evaluation:
>
>
> | Method           |      SC ↑ |      BC ↑ |      TF ↑ |      MS ↑ |     CTS ↑ |      WE ↓ |      IQ ↑ |      CS ↑ |
> | ---------------- | --------: | --------: | --------: | --------: | --------: | --------: | --------: | --------: |
> | FIFO             |     91.17 |     93.26 |     94.60 |     96.55 |     99.77 |     0.017 | **54.61** |     20.71 |
> | FIFO + Ours      | **92.64** | **94.26** | **96.75** | **98.06** | **99.90** | **0.007** |     54.27 | **21.27** |
> | FreeNoise        |     96.04 |     96.71 |     96.66 |     97.41 |     99.89 |     0.012 |     59.33 |     21.08 |
> | FreeNoise + Ours | **98.98** | **98.59** | **97.47** | **97.88** | **99.94** | **0.006** | **61.24** | **21.11** |
>
> These results show that our methods **consistently improve** the consistency upon existing methods on the VBench prompt set, with imaging and semantic quality **maintained**. Together, these results provide **stronger evidence for the effectiveness and generalizability** of our proposed methods across broader, more standardized prompt sets. We will revise our claims to more accurately reflect the scale of the original benchmark and incorporate these standardized evaluations to strengthen the empirical support for generalization.

---

> ### Author Response · Authors · 2026-06-28
>
> (Continue from the last response)
>
> ### 4. A human preference evaluation would strengthen the quality claims
>
> We conducted a **user preference study** to evaluate the generated single-prompt and multi-prompt videos based on human preferences. During evaluation, participants were explicitly instructed to focus on visual consistency across frames and overall video quality. As shown in the table, **users consistently preferred videos generated with our method over those generated without it, demonstrating its substantial improvement in user-perceived quality.** Specifically, our ViDE framework was applied to three baseline models for single-prompt generation and one baseline model for multi-prompt generation, achieving preference rates of over 76% and 87%, respectively.
>
> | Generation setting |            Baseline | With our method | Without our method |
> | ------------------ | ------------------: | --------------: | -----------------: |
> | Single-prompt      |                FIFO |      **76.49%** |             23.51% |
> | Single-prompt      | StreamingT2V (ST2V) |      **77.39%** |             22.61% |
> | Single-prompt      |           FreeNoise |      **77.26%** |             22.74% |
> | Multi-prompt       |                FIFO |      **87.89%** |             12.11% |
>
>
> ### 5. PromptBlend's LLM-based decomposition is insufficiently analyzed
>
> We thank the reviewer for this suggestion. In the revision, we will report the LLM identity, the complete decomposition instruction, inference cost, representative outputs, and common failure cases. We use ChatGPT to reorganize each prompt into five semantic slots: **Subject, Action, Place, Time, and Video Quality Description**. The LLM is instructed to preserve the original information, reorder it using “($)” separators, leave missing components empty, and make only minimal grammatical corrections.
>
> For example, “A fat rabbit wearing a purple robe walking through a fantasy landscape” is decomposed as
>
> `A fat rabbit wearing a purple robe $ walking $ through a fantasy landscape $ $.`
>
> These five components are **flexible semantic alignment slots** rather than a required structure for the original prompt. Free-form prompts can be mapped to the most relevant slots, missing components remain empty, and inseparable information can stay within one component. Typical failure cases include ambiguous slot boundaries, attribute assignment, and prompts with multiple subjects or actions. Since the instruction emphasizes extraction rather than content generation, such errors usually affect slot assignment rather than the underlying semantics.
>
>
>
> ### 6. The language around diagonal attention is too causal
>
> We agree and will revise the wording accordingly. The current observational results establish a statistical correlation between diagonal concentration and video inconsistency. The controlled intervention shows that reducing diagonal attention can improve consistency under the tested settings, but this does not establish diagonal attention as the sole or universal cause.
>
> We will therefore clearly distinguish the observational evidence from the intervention result. We will replace statements such as “diagonal attention causes video inconsistency” with “diagonal attention concentration is associated with video inconsistency, and reducing it improves consistency under the evaluated settings.”
>
> ### 7. Broader-impact discussion is missing
>
> ViDE enables more coherent long-video generation without additional training, which may support applications such as digital storytelling, education, and content creation. However, improved realism and consistency may also increase risks of misinformation, impersonation, and copyright misuse. The method further inherits biases and safety limitations from its underlying generative models. Responsible use should therefore include content disclosure, provenance or watermarking mechanisms, safety filtering, and human oversight. We will add these discussions to our paper.
>
> [1] Cai, M., Cun, X., Li, X., Liu, W., Zhang, Z., Zhang, Y., ... & Yue, X. (2025). Ditctrl: Exploring attention control in multi-modal diffusion transformer for tuning-free multi-prompt longer video generation. In Proceedings of the Computer Vision and Pattern Recognition Conference (pp. 7763-7772).

---

### Review · Reviewer_DmMP · 2026-06-03

**Summary Of Contributions:**

The paper proposes a tuning-free method for improving video coherence in long-video generation by optimizing the latent states (attention maps and feature embeddings). It mainly comprises two modules:
- Temporal attention reweighting, including:
	- attenuating the diagonal weights in the temporal attention matrix to improve video temporal coherence -- this is based on a key observation that the diagonal concentration of temporal attention is correlated to video inconsistency;
	- a time-frequency analysis for motion intensity for adaptively adjusting diagonal attention weight while avoiding blurriness.
- Prompt blending, including: LLM-based prompt reorganization, content alignment, and token interpolation.

**Audience:**

Yes

**Audience Explanation:**

I think the content in the paper would be interesting to the community in a few aspects:
- Long video generation is an important problem, and currently, it's relatively less explored than short video generation. However, one concern here is that, the generated videos in this paper are mostly ~6s, and I don't think they count as "long videos"?
- The obervation of video consistency v.s. temporal attention matrix shape, with strong and clear evidences, is interesting.
- The method doesn't require any data-specific finetuning, so it's general to some extent (also as demonstrated in the experiments, it can be applied different base architectures). I somehow understand it as a "smoothness regularization term" -- but not in a brute-force way like a pixel-wise L2 or optical flow regularization, instead, it is designed on the latent features and by observing their relations to video content consistency, which I think is both natural and smartly designed.

**Broader Impact Concerns:**

None.

**Claims And Evidence:**

Yes

**Claims Explanation:**

- I like the analysis of video consistency v.s. temporal attention diagonal value concentration. When reading the first few pages of the paper, I felt the method and results sounds a bit "magical", with two major questions:
	- (i) Are there clear evidences that the temporal attention has causation/correlation to the video inconsistency? -- intuitively, it makes some sense, but as this is the main motivation for the proposed method, we'll need strong evidences.
	- (ii) Simply applying the reweighting can make it work? -- without any finetuning of the model weights, but it doesn't cause any "out-of-distribution" failures?
I think the analysis conducted in the paper answers these questions quite well, particularly question (i). I appreciate the authors for giving both concrete examples and statistical analysis to best explain the phenomena.

- For the theoretical analysis in the paper, I also like it that: (i) it makes a clear list of assumptions, and discusses the validity of these assumptions; (ii) it shows how the inconsistency in video-frame pixel values relates to the temporal attention and the reweighting.

- For the experiments:
	- The quantitative and qualitative results show the effectiveness of the proposed method.
	- The ablation studies show the effectiveness of each component of the method.
	- However, I do have a question/concern here: The paper adopts the VBench metrics, but not their prompts -- instead, it proposes its own prompt set. I looked at the single prompt list in the appendix, and the sentences are quite short. I think the VBench prompts are much longer and also more context-rich, and they may be a better setup for long-video generation? Plus that it is an established public benchmark, testing with their prompts would be more convincing?

**Requested Changes:**

- I mainly want to see either (i) experimental results on VBench with their prompts, or (ii) explanations of why such an experiment is less appropriate or feasible.
- The generated videos in this paper are mostly ~6s, and I don't think they count as "long videos"?
- The attention matrices and analysis shown in the paper are all for 2D attention maps? How does things look like on 3D attention like CogVideoX?

---

> ### Author Response · Authors · 2026-06-28
>
> We thank the reviewer for recognizing the motivation, empirical evidence, theoretical formulation, and generality of our tuning-free method. We address the requested changes below.
>
> ### 1. The evaluation should use the official VBench prompts
>
> Thanks for the suggestion. We agree that evaluation on an established public prompt set would make the evidence more convincing. We therefore have added experiments using the official VBench prompts, where we measure the **consistency** (Subject Consistency (SC), Background Consistency (BC), Temporal Flickering (TF), Motion Smoothness (MS), CLIP-Temp Score (CTS), Warping Error (WE)) ,and **imaging and semantic quality** (Imaging Quality (IQ), and CLIP Score (CS)):
>
>
> | Method           |      SC ↑ |      BC ↑ |      TF ↑ |      MS ↑ |     CTS ↑ |      WE ↓ |      IQ ↑ |      CS ↑ |
> | ---------------- | --------: | --------: | --------: | --------: | --------: | --------: | --------: | --------: |
> | FIFO             |     91.17 |     93.26 |     94.60 |     96.55 |     99.77 |     0.017 | **54.61** |     20.71 |
> | FIFO + Ours      | **92.64** | **94.26** | **96.75** | **98.06** | **99.90** | **0.007** |     54.27 | **21.27** |
> | FreeNoise        |     96.04 |     96.71 |     96.66 |     97.41 |     99.89 |     0.012 |     59.33 |     21.08 |
> | FreeNoise + Ours | **98.98** | **98.59** | **97.47** | **97.88** | **99.94** | **0.006** | **61.24** | **21.11** |
>
> These results show that our methods **consistently improve** the consistency upon existing methods on the VBench prompt set, with imaging and semantic quality **maintained**.
>
>
> ### 2. Videos of approximately six seconds may not qualify as “long videos”
>
> We appreciate this question. Our base models typically generate 1~2 second clips, so generating 6+ seconds already requires temporal extension beyond their native horizon and can expose accumulated consistency issues. We use this duration for large-scale quantitative evaluation because substantially longer videos are prohibitively expensive to batch-test across multiple pipelines and ablations. Importantly, our multi-prompt experiments also include videos **longer than 30 and 50 seconds**, demonstrating the applicability of our method to substantially longer generations.
>
> ### 3. The attention analysis appears to focus on factorized 2D temporal attention; the extension to CogVideoX is unclear
>
> Empirically, our experiments on CogVideoX demonstrate that our methods are effective for 3D attention. In CogVideoX's full-3D attention, each spatiotemporal token jointly attends to tokens across both spatial and temporal dimensions. To analyze this attention structure, we group attention entries according to the temporal indices of the query and key tokens, allowing us to measure same-frame and cross-frame interactions while preserving spatial interactions within each temporal group. As shown in Table 1, our methods **remain effective** under this 3D attention setting.
>
> Conceptually, analyzing 3D attention along the temporal axis follows the same principle as our temporal-attention analysis in Section4. We will add a clearer visualization of the full-3D case to illustrate how the analysis and intervention extend to CogVideoX.

---

### Review · Reviewer_Zqep · 2026-06-15

**Summary Of Contributions:**

**Summary Of Contributions**

The paper addresses the problem of frames not staying consistent in training-free long video generation, which it attributes to two causes: the train–inference gap introduced by length-extension methods (FIFO, FreeNoise), and coarse interpolation in the multi-prompt setting. The proposed method, ViDE, has two components. TiTAR is based on the observation that inconsistent frames show diagonal-focused temporal attention and high-frequency energy in the DSTFT; it subtracts a value from the attention diagonal before softmax, chosen per frame from a DSTFT motion estimate so that fast motion is not over-smoothed, and provides a theorem stating that this reweighting reduces a frequency-based inconsistency measure. PromptBlend uses an LLM to split each prompt into five fixed components, aligns and interpolates them across prompts, and applies the interpolation gradually over denoising timesteps and U-Net layers. The method is evaluated on five pipelines using VBench/EvalCrafter metrics, along with several ablation tests.

**Strengths**
1. I like that it's a simple plug-and-play design, and the authors show it works across several length-extension methods and backbones, including full-3D attention (CogVideoX), rather than just a single setup.
2. The motivation is easy to follow—diagonal dominance plus high-frequency energy—and they support it with a theorem, which is fairly uncommon in this largely empirical area.
3. The ablations are solid: they separate out TiTAR, prompt alignment, and adaptive interpolation, and show that $\alpha$ is reasonably stable.

**Weaknesses**
1. My main concern is the evaluation. It relies entirely on automatic consistency metrics that are already very high, and these can be inflated simply by reducing motion. Without any human or preference study, I'm not convinced the gains reflect videos that actually look better.
2. The novelty feels not that significant to me. Attention reweighting is a well-established idea, frequency analysis in video diffusion has clear prior work (FreeInit, FreeLong), and PromptBlend reads more like a useful engineering recipe than a genuinely new method.
3. The theorem rests on fairly strong assumption, and it largely restates the starting assumption that inconsistency corresponds to high-frequency energy.
4. More comparison would be better. There is little direct comparison with other training-free consistency methods, the multi-prompt test set is small and authored by the authors themselves (13 sets), and the five-component prompt template is rigid, with no test on free-form prompts. There are also minor typo issues (e.g., "TiTAR" vs "TiARA" in the figures).

**Audience:**

Yes

**Audience Explanation:**

The paper should be of interest to researchers working on long video generation and training-free extension of pretrained video diffusion models. The observation linking temporal inconsistency to diagonal dominance in temporal attention and to high-frequency DSTFT energy is a useful diagnostic that others in this area could build on, and the proposed TiTAR module is simple, plug-and-play, and shown to work across several length-extension methods and backbones, including full-3D attention. The accompanying theoretical analysis, even if based on simplifying assumptions, is relatively uncommon in this largely empirical subfield and may interest readers who want a more principled view of frequency-based methods in video diffusion. PromptBlend and the structured prompt-alignment recipe would likewise be of practical interest to those handling multi-prompt transitions. Even though the gains are incremental and the backbones somewhat dated, the methods and findings are general enough that a meaningful subset of the community would find them worth knowing.

**Broader Impact Concerns:**

I have not found any discussions about the limitations and potential negative societal impact. But in my opinion, this may not be a problem, since the work focuses on the video generation tricks. Still, it is highly encouraged to add corresponding discussions.

**Claims And Evidence:**

Yes

**Claims Explanation:**

As scoped, the claims are clearly supported. The motivating observations are documented both qualitatively and statistically, and the main claim that TiTAR improves consistency holds across five pipelines with no real drop in imaging quality, generalizing even to full-3D attention (CogVideoX). The multi-prompt results and the ablations further isolate each component and show robustness to $\alpha$. The theorem, though based on simplifying assumptions, is consistent with the empirical findings.

My remaining concerns are about strengthening the evidence, not doubting it: a human/preference study would make the quality claim more convincing, and broader comparisons plus a larger, externally sourced multi-prompt test set would solidify the multi-prompt results.

**Requested Changes:**

See weakness.

---

> ### Author Response · Authors · 2026-06-28
>
> We thank the reviewer for recognizing ViDE's plug-and-play design, applicability across multiple architectures, clear motivation, and comprehensive ablations. We address the concerns below.
>
> ### 1. Automatic metrics may reward reduced motion rather than genuine visual improvement
>
> We agree that temporal-consistency metrics may favor overly static videos. TiTAR **avoids this by using DSTFT-based motion estimation** to preserve motion while improving cross-frame coherence.
>
> In addition to the automatic metrics, we also conducted a **user preference study** to evaluate the generated single-prompt and multi-prompt videos based on human preferences. During evaluation, participants were explicitly instructed to focus on visual consistency across frames and overall video quality. As shown in the table, users **consistently preferred videos generated with our method** over those generated without it, demonstrating its substantial improvement in user-perceived quality. Specifically, our ViDE framework was applied to three baseline models for single-prompt generation and one baseline model for multi-prompt generation, achieving preference rates of over 76% and 87%, respectively.
>
> | Generation setting |            Baseline | With our method | Without our method |
> | ------------------ | ------------------: | --------------: | -----------------: |
> | Single-prompt      |                FIFO |      **76.49%** |             23.51% |
> | Single-prompt      | StreamingT2V (ST2V) |      **77.39%** |             22.61% |
> | Single-prompt      |           FreeNoise |      **77.26%** |             22.74% |
> | Multi-prompt       |                FIFO |      **87.89%** |             12.11% |
>
> We further evaluate motion quality by jointly analyzing Warping Error (WE), which reflects temporal inconsistency, and Optical Flow Score (OFS), which measures motion intensity. The result is indicated in this anonymous [figure](https://ibb.co/m51fJtFC). We randomly sample $170$ videos from FIFO, FIFO+TiTAR, and UCF-101, and compute $\ln({WE})$ and $\ln(OFS)$ at an effective frame rate of $10$ FPS. The regression slope of our method is lower than that of FIFO and closer to that of real videos, indicating that it better controls temporal distortion as motion intensity increases. Moreover, the KL divergence between the joint WE--OFS distributions of our generated videos and real videos decreases from $1.13$ for FIFO to $0.82$ for FIFO+ViDE. These results suggest that our method produces **smoother and more realistic** motion patterns, which lead to **genuine visual improvement**.
>
>
> ### 2. Novelty Compared to Existing Frequency-Analysis Methods
>
> Existing frequency-based methods primarily manipulate the spectral components of latent features or initial noise to improve video quality. For example, FreeLong applies a global 3D Fourier transform and blends the low-frequency components of global-attention features with the high-frequency components of local-attention features, primarily addressing the frequency distortion caused by directly extending short-video models. **In contrast**, our TiTAR uses localized time-frequency analysis of temporal attention maps to distinguish normal motion from inconsistency and dynamically reweights attention according to spatially and temporally varying motion intensity. Thus, rather than directly filtering or combining latent-frequency components, our method uses frequency analysis as an **adaptive signal** for correcting the underlying attention behavior, making it **plug-and-play** across different long-video generation pipelines and architectures; we further provide a **theoretical analysis** connecting attention reweighting to reduced high-frequency inconsistency.
>
> TiTAR also outperforms FreeLong empirically, as detailed in our response to Question 4. This further highlights of the novelty of our method.
>
>
> ### 3. The theorem relies on strong assumptions and equates inconsistency with high-frequency energy
>
> Thanks for raising this question. Our assumption that visual inconsistency is associated with high-frequency energy is supported both **intuitively** and **empirically**.
>
> **Intuitively**, video inconsistency often manifests as temporal flickering, abrupt scene changes, or sudden changes in object appearance. These phenomenons are often associated with the high-frequency components. Empirically, we **verify this assumption in Figure 6**. Specifically, we classify video clips into three categories: (1) slow-motion clips, (2) intense-motion clips, and (3) inconsistent clips. The results show that **inconsistent clips contain substantially larger energy in the high-frequency components** than consistent clips, including those with either slow or intense motion. This suggests that the high-frequency signal captured by our method is not merely caused by fast motion, but is **closely related to temporal inconsistency**.
>
> We will add more explanations of this assumption in the revised version.

---

> ### Author Response · Authors · 2026-06-28
>
> (Continue from the last response)
>
> ### 4. The evaluation would benefit from broader and more standardized comparisons
>
> Thanks for the suggestion. We strengthen our evaluations from two perspectives.
>
> First, we add **experiments using the official VBench prompts** across representative pipelines, and we measure the **consistency** (Subject Consistency (SC), Background Consistency (BC), Temporal Flickering (TF), Motion Smoothness (MS), CLIP-Temp Score (CTS), Warping Error (WE)) ,and **imaging and semantic quality** (Imaging Quality (IQ), and CLIP Score (CS)):
>
>
> | Method           |      SC ↑ |      BC ↑ |      TF ↑ |      MS ↑ |     CTS ↑ |      WE ↓ |      IQ ↑ |      CS ↑ |
> | ---------------- | --------: | --------: | --------: | --------: | --------: | --------: | --------: | --------: |
> | FIFO             |     91.17 |     93.26 |     94.60 |     96.55 |     99.77 |     0.017 | **54.61** |     20.71 |
> | FIFO + Ours      | **92.64** | **94.26** | **96.75** | **98.06** | **99.90** | **0.007** |     54.27 | **21.27** |
> | FreeNoise        |     96.04 |     96.71 |     96.66 |     97.41 |     99.89 |     0.012 |     59.33 |     21.08 |
> | FreeNoise + Ours | **98.98** | **98.59** | **97.47** | **97.88** | **99.94** | **0.006** | **61.24** | **21.11** |
>
>
> These results show that our methods **consistently improve** the consistency upon existing methods on the VBench prompt set, with imaging and semantic quality **maintained**.
>
> Second, we also **include FreeLong as a closely related training-free, frequency-based baseline** under the same prompt set and evaluation protocol:
>
> | Method   |      SC ↑ |      BC ↑ |      TF ↑ |      MS ↑ |     CTS ↑ |      WE ↓ |      IQ ↑ |      CS ↑ |
> | -------- | --------: | --------: | --------: | --------: | --------: | --------: | --------: | --------: |
> | FreeLong |     92.75 |     95.54 |     96.79 |     97.71 |     99.89 |     0.013 |     52.91 |     20.85 |
> | Ours     | **98.98** | **98.59** | **97.47** | **97.88** | **99.94** | **0.006** | **61.24** | **21.11** |
>
> These additional experiments **further validate the effectiveness of our method** compared with existing methods. They provide a more standardized evaluation, and better position TiTAR relative to existing training-free consistency methods.
>
> We will add these results and discussions in the revision.
>
> ### 5. The multi-prompt benchmark is small and author-constructed
>
> We strengthen the multi-prompt evaluation by expanding the evaluation prompt set. Specifically, we will additionally evaluate FIFO and FIFO+ViDE using the multi-prompt set independently introduced by DiTCtrl[1].
>
> | Method      |  SC ↑ |  BC ↑ |  TF ↑ |  MS ↑ | CTS ↑ |  WE ↓ |  IQ ↑ |  CS ↑ |
> | ----------- | ----: | ----: | ----: | ----: | ----: | ----: | ----: | ----: |
> | FIFO        | 79.30 | 89.40 | 94.02 | 96.47 | 99.65 | 0.019 | **52.77** | 18.93 |
> | FIFO + Ours | **82.42** | **90.32** | **96.10** | **98.10** | **99.78** | **0.007** | 52.14 | **19.03** |
>
> The results show that our methods **consistently achieve better performance** on this additional evaluation prompt set.
>
>
> ### 6. PromptBlend may depend on a rigid five-component template
>
> The five semantic components are **flexible alignment slots** rather than a required input format. Users may provide free-form prompts, which are then reorganized by an LLM into the available slots. Missing components may remain empty or unchanged, while information that cannot be cleanly separated can be retained within a single component.
>
> ### 7. Causal claims about diagonal attention should be softened
>
> We agree that the motivation analysis establishes correlation rather than causation. We will revise the wording throughout the paper to clearly distinguish the observed correlation, the controlled intervention, and the resulting empirical improvements.
>
> ### 8. Typos, limitations, and broader impact
>
> ViDE enables more coherent long-video generation without additional training, which may support applications such as digital storytelling, education, and content creation. However, improved realism and consistency may also increase risks of misinformation, impersonation, and copyright misuse. The method further inherits biases and safety limitations from its underlying generative models. Responsible use should therefore include content disclosure, provenance or watermarking mechanisms, safety filtering, and human oversight. We will correct all the typos and add these discussions to our paper.
>
> [1] Cai, M., Cun, X., Li, X., Liu, W., Zhang, Z., Zhang, Y., ... & Yue, X. (2025). Ditctrl: Exploring attention control in multi-modal diffusion transformer for tuning-free multi-prompt longer video generation. In Proceedings of the Computer Vision and Pattern Recognition Conference (pp. 7763-7772).

---

### Author Response · Authors · 2026-07-19

Dear AC and Reviewers,

We have revised the manuscript in accordance with your valuable feedback where the changes are highlighted in red. The revised version has been uploaded for your review. We would greatly appreciate it if you could kindly check the updated manuscript. If you have any further questions or comments, we would be pleased to address them.

We would like to express our gratitude for your time and efforts during the review process.

Sincerely,
Authors